# Morphodynamics of surface-attached active drops

Alejandro Martínez-Calvo [1,2,3] ✉ & Sujit S. Datta [2,4] ✉

Many biological and synthetic systems are suspensions of oriented actively-moving components. Unlike in passive suspensions, the interplay between orientational order, active flows, and interactions with boundaries gives rise to fascinating new phenomena in such active suspensions. Here, we examine the paradigmatic example of a surface-attached drop of an active fluid (an "active drop"), which has so far only been studied in the idealized limit of thin drops. We find that such surface-attached active drops can exhibit a wide array of stable steady-state shapes and internal flows that are far richer than those documented previously, depending on boundary conditions and the strength of active stresses. Our analysis uncovers quantitative principles to predict and even rationally control the conditions under which these different states arise —yielding design principles for next-generation active materials.

Active matter refers to systems whose components consume energy from their environment and convert it into motion and/or biomass, thereby driving them far from equilibrium[1–3]. Many biological and synthetic active systems are compartmentalized and therefore resemble liquid drops, with an interface that separates a suspension of active units from the surrounding environment. These units can self-organize and acquire orientational order, exert stresses, and induce spontaneous flows that can deform the confining interface. In turn, these changes in shape can influence the organization of the active units. This interplay between order, flow, and shape plays a key role in the function of diverse forms of active matter, such as organoids[4,5], bacterial and amoeba colonies[6–12], the mitotic spindle[13–15], microtubule vesicles[16], droplets in active suspensions[17,18], phase-separated fibril drops[19], intracellular and developmental flows[20–23], and colloidal and ferrofluid droplets[24–26].

In both natural and synthetic contexts, active drops do not typically exist in isolation, but are attached to surfaces, such as in microbial colonies[6–8,10–12], single-cell and tissue migration[27–31], and active wetting phenomena[24–26,29,32,33]. Understanding the morphodynamics of these surface-attached active systems is of fundamental interest in active matter physics and has key implications for both biology and materials science. How does the coupling between orientational order, flow, and interactions with confining interfaces influence the shape of surface-attached active drops? A comprehensive answer to this question is still

missing, despite the critical role of this interplay in the functioning of active systems. In particular, while current theoretical models provide useful intuition, they rely on the thin-film approximation[12,34–41], which assumes that all variables describing the drop vary less in the direction parallel to the substrate than in the direction normal to it[42,43]. This strong approximation is not applicable to many real-world systems, for which the drop shape, ordering of active units, and self-generated flows can vary in all directions. As a result, current understanding is incomplete. Indeed, by relaxing the thin-film approximation, here, we show that the morphodynamics of surface-attached drops are far richer than was previously known.

We revisit the paradigmatic continuum model of an active drop– containing a suspension of active units that exhibit nematic order and spontaneously self-generate flow–attached to a rigid, impermeable, flat surface. Unlike previous work, we do not make the thin-film approximation, but instead perform time-dependent simulations of the complete conservation equations. We find that the drop can exhibit a rich array of steady-state shapes and internal flows, which are a function of the alignment of active units with the boundaries and the active Capillary number comparing the active stresses exerted by the units and the capillary pressure. We uncover all possible states of the drop and the necessary conditions for symmetry breaking, which only emerges spontaneously when (i) the units become polarized at the solid substrate and the liquid–air interface of the drop and (ii) when

[1]Princeton Center for Theoretical Science, Princeton University, Princeton, NJ, USA. [2]Department of Chemical and Biological Engineering, Princeton University, Princeton, NJ, USA. [3]Department of Physics, Princeton University, Princeton, NJ, USA. [4]Present address: Division of Chemistry and Chemical Engineering, California Institute of Technology, Pasadena, CA, USA. ✉e-mail: amcalvo@princeton.edu; ssdatta@caltech.edu

they adopt a particular alignment direction at those boundaries. These results are in stark contrast to those generated using the thin-film approximation, which predicts a unique equilibrium shape for a given absolute value of the Capillary number, independent of alignment conditions and the contractile or extensile nature of the active stresses. Finally, we show that these equilibrium drop shapes can be controlled reversibly via surface anchoring and through the bulk active stresses exerted by the units.

Altogether, our findings provide a deeper understanding of the morphodynamics of active drops, with implications for various biological processes such as cell migration, tissue morphogenesis, biofilm development, and intracellular flows. These insights not only shed light on the complex interplay between order, flow, and shape in living systems but also offer a framework for the design and control of synthetic active materials and living systems.

## Results

### Model of an active nematic drop attached to a solid substrate

We investigate the morphodynamics of surface-attached viscous nematic droplets: drops of fluid containing a suspension of elongated active units that acquire orientational order with head-tail symmetry. These units tend to maintain such order and exert force dipoles that spontaneously generate flow, thereby deforming the interface between the active fluid inside the drop and the surrounding fluid, which we consider to be passive and quiescent (Fig. 1a). For simplicity, we consider a two-dimensional (2D) continuum model to describe a nematic droplet of radius $R$ of an incompressible, Newtonian fluid of dynamic viscosity $\mu$ and surface tension $\gamma$ associated with the liquid−air interface. We assume that the droplet is pinned to a rigid, impermeable substrate extending from $0 \leq x \leq 2R$, with an initial semicircular shape, and contains a uniform suspension of active units. The nematic orientational order of the units is described by the director vector field $\boldsymbol{p}(\boldsymbol{r}, t)$, where $\boldsymbol{r}$ is position and $t$ is time. In this description, the active units can polarize at the boundaries of the drop (see Fig. 1; and Supplementary Information for a full nematic description) The flow generated by the active stress exerted by the units is described by the fluid isotropic pressure $\Pi(\boldsymbol{r}, t)$ and the velocity field $\boldsymbol{u}(\boldsymbol{r}, t)$, assuming Stokes flow, i.e., low-Reynolds-number flow. The dimensionless volume and momentum conservation equations read:

$$\nabla \cdot \boldsymbol{u} = 0, \text{ and } \boldsymbol{0} = \nabla \cdot \boldsymbol{\sigma}, \quad (1)$$

where $\boldsymbol{\sigma}$ is the stress tensor of the fluid. For simplicity, we consider that active units do not proliferate. The stress tensor takes into account the fluid isotropic pressure $\Pi$, which enforces flow incompressibility (Eq. (1)), the viscous stress, and the active stress generated by the active components in the drop. Assuming that the flow is Newtonian and neglecting any complex rheology of the fluid resulting from the presence of the active units, the stress tensor is given by:

$$\boldsymbol{\sigma} = -\Pi I + \mu[\nabla \boldsymbol{u} + (\nabla \boldsymbol{u})^T] - \alpha \boldsymbol{p}\boldsymbol{p}, \quad (2)$$

where $\alpha$ is the signature of the active stress, which quantifies the strength of the force dipoles exerted by the active units. The sign of $\alpha$ characterizes the nature of the flow produced by the active units: for $\alpha > 0$, the flow is *extensile*, while for $\alpha < 0$, it is *contractile*, favoring stretching or contraction along the axis of the active unit, respectively. In a biological context, swimming microorganisms can produce either contractile or extensile flow depending on their motility mode[1]. Similarly, microtubule suspensions can exhibit either extensile or contractile behavior[16,44–46].

To determine the orientation field $\boldsymbol{p}$ describing the nematic order of active units in the drop, we consider the following dynamical equation, $\Gamma(\partial_t \boldsymbol{p} + \boldsymbol{u} \cdot \nabla \boldsymbol{p} + \Omega \cdot \boldsymbol{p}) = -\delta F/\delta \boldsymbol{p}$, where $\Gamma$ is a rotational viscosity, $\Omega$ is the vorticity tensor, and the right-hand side corresponds to the first variation of the free-energy functional $F$ with respect to $\boldsymbol{p}$[1,47]. For simplicity, we assume that this order is established much more quickly than the flows generated within the drop, which yields $\delta F/\delta \boldsymbol{p} = \boldsymbol{0}$. This assumption implies that the orientation field relaxes instantaneously toward its free-energy minimum and is uncoupled from the flow dynamics, usually referred to as the strong elastic limit (see Supplementary Information). We consider the following free energy for the orientation field[48–50]:

$$F = \int_A \mathrm{d}A \frac{K}{2}(\nabla \boldsymbol{p} : \nabla \boldsymbol{p}), \quad (3)$$

where $K$ is the Frank elastic constant in the one-constant approximation[48,49] and $A$ is the area of the 2D drop. The first variation of $F$ with respect to $\boldsymbol{p}$ yields the following equation for the instantaneous orientation field,

$$\nabla^2 \boldsymbol{p} = \boldsymbol{0}. \quad (4)$$

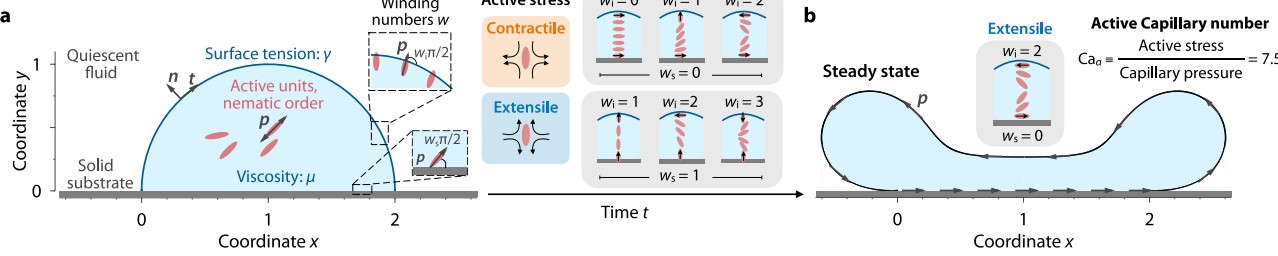

**Fig. 1 | Surface-attached active drops adopt stable, steady-state shapes and internal flows. a** Initial shape of a 2D liquid drop of viscosity $\mu$ and surface tension $\gamma$, attached to a rigid, impermeable substrate. The drop contains a uniform suspension of active units that exhibit nematic orientational order and self-generate flow through active stresses. These stresses can either be contractile, producing compressive flow along the axis of the units, or extensile, resulting in elongational flow. The vector field $\boldsymbol{p}(\boldsymbol{r}, t)$ characterizes the orientational order of the units. These units can polarize at the boundaries, forming an angle of $w_s\pi/2$ with the solid substrate and an angle of $w_i\pi/2$ with the liquid−air interface. Here, $w$ denotes the winding number, which represents the number of quarter turns the units make with respect to the direction tangential to the boundaries. The possible anchoring configurations include: $w_s = 0$ and $w_i = 0, 1, 2$ for planar substrate anchoring, and $w_s = 1$ and $w_i = 1, 2, 3$ for orthogonal (homeotropic) substrate anchoring. **b** Stresses induced by activity spontaneously generate flow, causing deformation of the drop. Over time, the surface-attached active drop reaches a steady-state shape and flow, where active, viscous, and surface tension forces are in balance. This state depends on three dimensionless parameters: (i) the active Capillary number $Ca_\alpha$, comparing the active stresses, which tend to deform the drop, with the capillary pressure, which tends to maintain a semicircular shape by minimizing the surface area per unit length; and (ii, iii) the orientation of the units, characterized by the winding numbers $w_s$ and $w_i$. Here, $Ca_\alpha = 7.5$, $w_s = 0$, and $w_i = 2$, implying that the flow is extensile, and the units are tangentially aligned with both boundaries, making two quarter turns from the substrate to the liquid−air interface (Supplementary Movie 1).

At the interface of the drop we impose a kinematic condition specifying that the velocity of the interface is equal to the velocity of the fluid, thus precluding mass transfer across the interface, the stress balance, and the orientation of the units:

$$\text{Fluid}:$$
$$(\partial_t \boldsymbol{r}_i - \boldsymbol{u}) \cdot \widehat{\boldsymbol{n}} = 0, \quad \boldsymbol{\sigma} \cdot \widehat{\boldsymbol{n}} = -\gamma \mathcal{C} \widehat{\boldsymbol{n}} \text{ at } \boldsymbol{r} = \boldsymbol{r}_i, \tag{5a}$$

$$\text{Interfacial polarization of active units}:$$
$$\boldsymbol{p} = \boldsymbol{R}_i \cdot \widehat{\boldsymbol{t}} \text{ at } \boldsymbol{r} = \boldsymbol{r}_i, \tag{5b}$$

where $\boldsymbol{r}_i$ is the position of the interface, $\widehat{\boldsymbol{n}}$ and $\widehat{\boldsymbol{t}}$ are the unit normal and tangential vectors to the interface, respectively, and $\mathcal{C} \equiv \boldsymbol{\nabla} \cdot \widehat{\boldsymbol{n}}$ is twice the mean curvature of the liquid–air interface. It is important to emphasize that while the active units exhibit nematic order in the bulk, satisfying the $\boldsymbol{p} \rightarrow -\boldsymbol{p}$ symmetry, they become polarized at the boundaries of the drop, as described by Eq. (5b) for the liquid–air interface (see Supplementary Information for a full nematic description of the active drop). Here, $\boldsymbol{R}_i$ is the rotation matrix prescribing the angle $\theta_i$ that the orientation of the units form with the tangent vector to the interface. For simplicity, we follow refs. 36,38 and consider only quarter turns in the orientation angle, i.e., $\theta_i = w_i \pi/2$, where $w \in \mathbb{Z}$ is the winding number (Fig. 1a).

At the solid substrate, we impose no-permeation and no-slip conditions for the velocity field, i.e., $\boldsymbol{u} = \boldsymbol{0}$ at $y = 0$, and we prescribe the polar orientation of the active units with respect to the tangent vector to the substrate, equivalently to the condition at the interface, i.e., $\boldsymbol{p} = \boldsymbol{R}_s \cdot \widehat{\boldsymbol{e}}_x$, where $\boldsymbol{R}_s$ is the corresponding rotation matrix prescribing the angle $\theta_s$ at the substrate. For simplicity, we also assume that the orientation angle with the substrate only changes in quarter turns, and thus it is specified by an additional winding number $w_s$. As initial conditions, we consider the shape of the drop to be semicircular and the fluid in the drop is at rest, i.e., $\boldsymbol{u}(\boldsymbol{r}, t = 0) = \boldsymbol{0}$ and $\Pi(\boldsymbol{r}, t = 0) = \gamma/R$.

## Dimensionless parameters governing the morphodynamics of a surface-attached active drop

Before solving Eqs. (1–5), we non-dimensionalize these equations to reduce the number of parameters. To this end, we choose the initial drop radius $R$ as the characteristic length scale, viscocapillary time $\mu R/\gamma$ as the characteristic time scale, $\gamma/\mu$ as the corresponding characteristic velocity scale, and capillary pressure $\gamma/R$ as the characteristic pressure scale. All results and variables hereafter are non-dimensionalized by these quantities. For simplicity, we retain the same notation for the dimensionless variables. Non-dimensionalization reveals only one governing dimensionless parameter, the *active Capillary number*:

$$Ca_\alpha \equiv \frac{\alpha}{\gamma/R} : \frac{\text{Active stress}}{\text{Capillary pressure}}, \tag{6}$$

which compares the active stress exerted by the active units with the capillary pressure[36,51]. Thus, our minimal model is described by three dimensionless parameters: the active Capillary number, $Ca_\alpha$, and the two winding numbers $w_i$ and $w_s$, which specify the orientation angles of the active units with respect to the liquid–air interface and solid substrate, respectively.

Previous works described surface-attached nematic drops using the thin-film approximation. This approximation simplifies Eqs. (1–5) into a single equation describing the height of the drop $h(\boldsymbol{r}, t)$ by assuming that all variables change slowly along the direction of the substrate compared to the variation in the direction normal to the substrate: $\partial_t h + \boldsymbol{\nabla} \cdot (h^3 \boldsymbol{\nabla} \nabla^2 h/3 + Ca_{\alpha,\text{lub}} h^2 \boldsymbol{r})$. This approach reduces the number of dimensionless parameters to one, which is a modified active Capillary number that absorbs a winding number $w$, determining the

orientational turns of the active units from the solid substrate to the liquid–air interface, $Ca_{\alpha,\text{lub}} \equiv \alpha/(2\pi w \gamma/R)$. Here, by contrast, we do not a priori impose the slenderness of the drop; instead, we conduct full numerical simulations of the conservation Eqs. (1–5).

## Surface-attached active drops reach stable steady states

How is the organization of active nematic units influenced by, and in turn influences the interfacial morphodynamics of a surface-attached nematic drop? To address this question, we perform numerical simulations of Eqs. (1–5). We first explore the case shown in Fig. 1b as an illustrative example, which corresponds to $Ca_\alpha = 7.5$, $w_s = 0$, and $w_i = 2$ (Supplementary Movie 1). In this case, the active units generate extensile flows and their orientation rotates by $\pi$ radians from the substrate to the liquid–air interface. This spatial organization of the active units, combined with the high active stresses they exert, generates strong flows capable of inducing large deformations of the drop. However, after a transient period, the drop eventually reaches a stable, steady-state shape and internal flow, where active, viscous, and capillary forces are balanced, as shown in the snapshot in Fig. 1b. This equilibrium shape and flow contrasts with the thin-film prediction[36,38]. Under this approximation, the active droplet always breaks symmetry, regardless of the value of the slender active Capillary number $Ca_{\alpha,\text{lub}}$ or the orientation of the units relative to the substrate and liquid–air interface.

Motivated by this finding, we next explore whether a surface-attached nematic drop always reaches a stable steady state for all values of the governing parameters, and investigate the necessary conditions for symmetry breaking, which can ultimately lead to drop migration.

## Surface-attached active drops adopt a rich array of equilibrium shapes and flows

What are the possible shapes that a surface-attached active drop can adopt? To address this question, we perform time-dependent simulations for all possible combinations of winding numbers $w_i$ and $w_s$, extensile and contractile stresses, and values of the active Capillary number $|Ca_\alpha|$. The possible anchoring configurations are $w_s = 0$ and $w_i = \{0, 1, 2\}$ for planar substrate anchoring, and $w_s = 1$ and $w_i = \{1, 2, 3\}$ for homeotropic substrate anchoring, as shown in the schematic in Fig. 1. We find that the drop adopts equilibrium shapes for all values of $Ca_\alpha$ and $w$ (Supplementary Movies 2-13). This result contrasts with previous works on active nematic drops that are not attached to a solid boundary, which have shown that such free drops undergo fingering instabilities for different anchoring conditions and values of $Ca_\alpha$[34,52] (Supplementary Movies 16 and 15). Moreover, the drop exhibits a wide variety of stable steady states, illustrated in the $(Ca_\alpha, w)$ state diagram in Fig. 2. Among these, some states spontaneously break symmetry. What conditions are necessary for this symmetry breaking?

## Parallel substrate anchoring

We begin by analyzing cases where the units align parallel to the solid substrate, corresponding to $w_s = 0$ (Fig. 2a). Additionally, we consider the case of $w_i = 0$, where the units are also aligned parallel to the liquid–air interface; our results are shown in the bottom row of Fig. 2a and Supplementary Movies 2–5). These anchoring conditions generate a bipolar structure in the ordering of units, with two point defects at the triple contact points between the solid substrate, the fluid, and the surrounding environment. Consequently, the order of active units, the flow, and the shape of the drop are symmetric with respect to its central midplane. Interestingly, for $-4.5 \lesssim Ca_\alpha \lesssim 2.5$, this nematic ordering produces two steady-state counter-rotating vortices, whose handedness is independent of the extensile or contractile nature of the active stresses. However, the drop shape is markedly different between the contractile ($Ca_\alpha < 0$) and extensile ($Ca_\alpha > 0$) cases. In the former, the drop contracts in the direction parallel to the

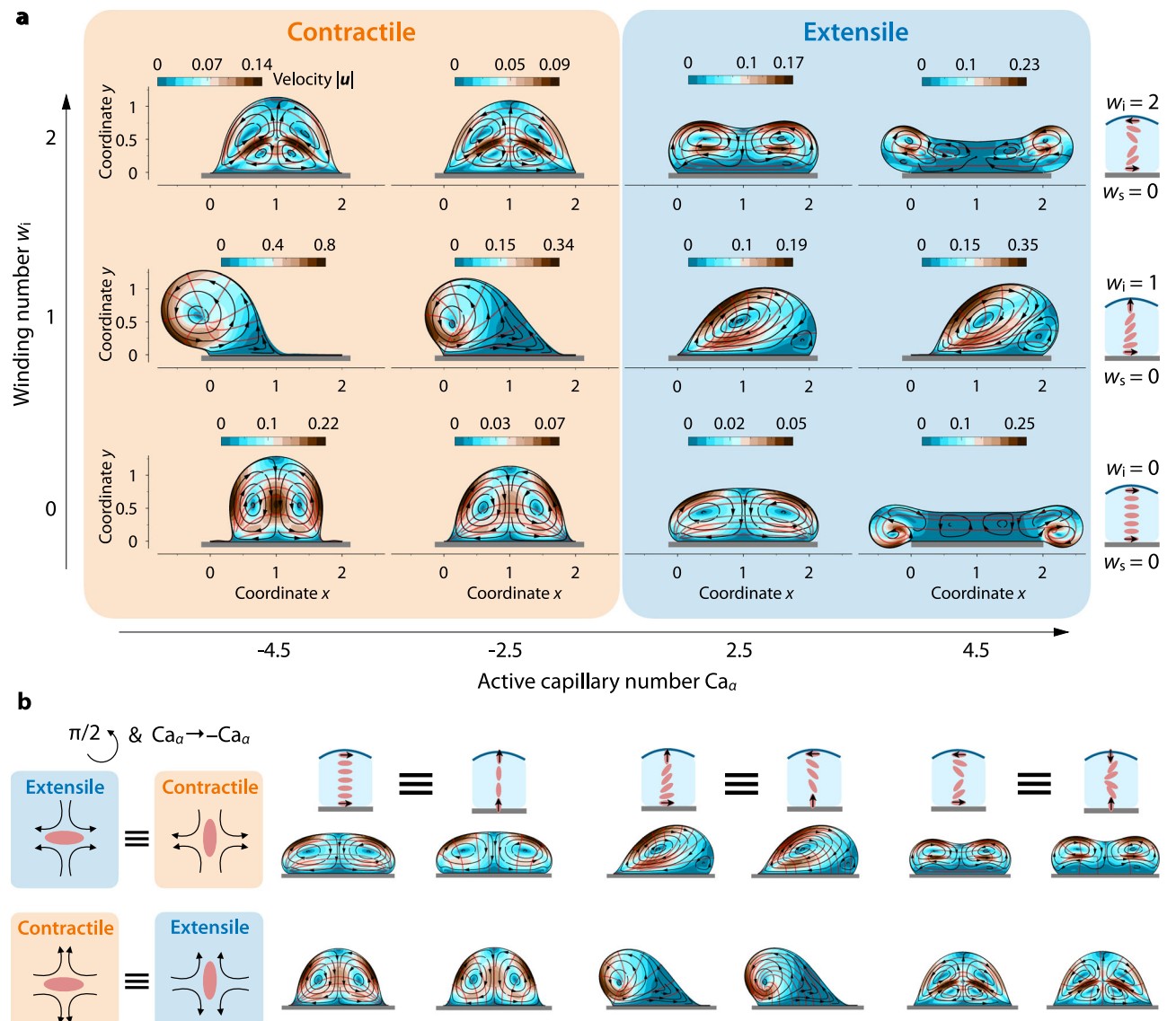

**Fig. 2 | Surface-attached active drops acquire a rich array of equilibrium shapes and flows. a** Steady-state shapes and self-generated flows of a 2D active nematic drop attached to a rigid, planar substrate, as a function of the active capillary number $Ca_\alpha$ and the interfacial winding number $w_i$, for planar substrate anchoring, $w_s = 0$ (Supplementary Movies 2 and 13). The color plots display the magnitude of the flow velocity $|u|$ inside the drop. The black curves represent the flow streamlines, and the red curves depict the orientation of the active units. **b** The steady states of a surface-attached active drop for planar ($w_s = 0$) and orthogonal ($w_s = 1$) unit anchoring with the substrate are equivalent under the transformations $w \to w + \pi/2$ and $Ca_\alpha \to -Ca_\alpha$. Here, $|Ca_\alpha| = 2.5$.

substrate, resulting in a *mushroom-like* shape. In the latter, the extensile stresses tend to flatten the drop along the $x$ direction. Intriguingly, for $Ca_\alpha \gtrsim 4.5$, the extensile stresses become strong enough to elongate and shrink the drop at its center, generating a flat film that connects two lobes with multiple *counter-rotating vortices*. This shape transition coincides with the emergence of a spiral defect inside the two lobes. These results contrast with the thin-film prediction, in which the condition $w_s = w_i = 0$ does not generate flow, and the drops remain undeformed. This is a consequence of neglecting the variation of **p** along the direction of the substrate.

We next focus on the case with $w_s = 0$ and $w_i = 1$, in which the units make a counterclockwise quarter turn from the substrate to the liquid–air interface (Fig. 2a, middle row and Supplementary Movies 6–9). Our results reveal that a mismatch in the orientation of boundary-polarized units is a necessary condition for symmetry breaking. The direction in which the drop breaks symmetry depends on the handedness of the units' rotation and the extensile and contractile nature of the active stresses. If the units make a clockwise rotation ($w_s = 0$ and $w_i = 3$), the shape and flow of the drop are identical to those in the case of counterclockwise rotation in Fig. 2a under a $x \to -x$ mirror symmetry. Furthermore, for the same units' rotation, the extensile and contractile cases lead to opposite flow vortex rotations. As a consequence, symmetry is broken in opposite directions. Intriguingly, the hedgehog defect in the units' ordering at one of the poles of the drop, observed in the extensile case, does not emerge in the contractile case, where instead a stable spiral defect appears in the bulk of the drop. Again, the asymmetric steady-state shapes and flows obtained here are markedly different from the thin-film prediction. Under this approximation, the extensile and contractile cases are identical under a $x \to -x$ reflection, and changing the handedness of the units' rotation is equivalent to changing the sign of $Ca_{\alpha,lub}$, which does not hold in the complete model.

Next, we explore the case of $w_s = 0$ and $w_i = 2$ (Fig. 2, top row and Supplementary Movies 10–13), where the units rotate $\pi$ radians from the substrate, where they are horizontally aligned, to the interface, where they are also tangentially aligned. As expected, this

configuration does not break the symmetry of the drop, unlike the thin-film equation. In particular, these anchoring conditions generate a vortex defect in the ordering of active units. In contractile cases, this defect generates a flow with two pairs of counter-rotating vortices, and the drop achieves a mushroom-like shape, similar to the case with $w_i = w_s = 0$. In extensile cases, the drop becomes more elongated, and high activity turns the vortex defect into a line defect, resulting in a complex flow structure with multiple vortices, similar to the case with $w_i = w_s = 0$.

### Homeotropic substrate anchoring

Thus far, we have explored cases in which the active units are aligned tangentially with the solid substrate ($w_s = 0$). To investigate whether other substrate anchoring configurations can alter the drop's morphodynamics, we conduct numerical simulations of drops in which the units are oriented orthogonally to the substrate ($w_s = 1$). We find that the drop shapes obtained with ($w_s$, $w_i$) = {(0, 0), (0, 1), (0, 2)} are equivalent to those obtained by a $\pi/2$ rotation of the units, i.e., ($w_s$, $w_i$) = {(1, 1), (1, 2), (1, 3)}, under the transformation $Ca_\alpha \rightarrow - Ca_\alpha$. This result makes sense intuitively, as the flow generated by an extensile unit oriented horizontally is equivalent to the one generated by a contractile unit oriented vertically, and vice versa (Fig. 1b). This finding is unlikely to hold if the strong elastic limit is relaxed, i.e., if the flow is coupled back with the orientation of the active units. Exploring the influence of this additional complexity will be a useful direction for future work.

### Describing the nematic field using a nematic tensor

Describing a nematic through a director field $p$ is not always equivalent to describing it through a nematic tensor $Q$. In our case, the key difference between the two descriptions arises from the boundary conditions: in the $p$-based formulation, the imposed boundary conditions do not respect the nematic symmetry ($p \rightarrow - p$), causing the active units to become polarized at the substrate and liquid–air interfaces. To understand the implications of these differences in our surface-attached drop, we describe the nematic through the following dynamical equation for Q, $\Gamma[\partial_t Q + u \cdot \nabla Q - (Q \cdot \Omega - \Omega \cdot Q)] = - \delta F_Q / \delta Q$, where $\Gamma$ is a rotational viscosity, and $F_Q$ a free-energy functional that reads[1,53–56]

$$F_Q = \int_A dA - \frac{a}{2} Q : Q + \frac{b}{4} (Q : Q)^2 + \frac{K_Q}{2} (\nabla Q)^2, \quad (7)$$

where $Q = S(pp - \frac{1}{2}I)$, with $S$ a scalar representing the strength of the order parameter, $K_Q$ the Frank elastic constant, and $a$ and $b$ parameters from a Landau expansion in the order parameter that determine the isotropic-nematic transition ($a > 0$ in the nematic phase and $b > 0$ for stability). These parameters define the maximum value the order parameter can take, $S_{max} = \sqrt{2a/b}$. Here, we also assume purely relaxational dynamics, i.e., the nematic relaxation rate is much faster than the flow strain rates (see Supplementary Information), which yields,

$$0 = [a - b \, tr(Q^2)]Q + K_Q \nabla^2 Q. \quad (8)$$

We numerically solve Eq. (8) together with Eqs. (1), where the active stress tensor is now given by $\sigma_a = - \alpha Q$ (see Supplementary Information)[1,53–57]. A key difference between the two frameworks is that the configurations ($w_s$, $w_i$) = (0, 0) and (0, 2) are identical for a true nematic described by $Q$ (Fig. S1a and Supplementary Information), whereas they are distinct when the nematic is described by the orientational vector field $p$ (Fig. 2a). Furthermore, symmetry breaking of the drop shape emerges only in the $p$ description, where the units become polarized at the drop boundaries, while it remains symmetric in the full nematic $Q$-tensor description. Nonetheless, the physical

principles and transformations under which the shape and hydrodynamics of parallel and orthogonal anchoring are equivalent remain the same for both descriptions (Figs. 2b and S1b and Supplementary Information). Moreover, in the Supplementary Information, we explore whether nematic advection plays a relevant role in the shape morphodynamics for both the $p$ and $Q$ descriptions—that is, when the nematic relaxation timescale toward the free-energy minimum is comparable to the flow timescale. While the transient dynamics with nematic advection differ from those in the strong-elastic limit, the final steady-state shapes and flows that the drop adopts are nearly identical in both cases (Figs. S2 and S3).

### Flow and energy dissipation in active drops follow simple scaling laws

Despite the complexity of shapes and internal flows exhibited by a surface-attached active nematic drop, steady-state flow inside the drop follows simple scaling laws (Fig. 3). In particular, the maximum velocity, $max(|u|) \equiv |u|_{max}$, follows identical linear scaling laws for both contractile and extensile cases when $|Ca_\alpha| \lesssim 1$, i.e., $|u|_{max} \sim |Ca_\alpha|$, for any value of $w_i$ (Fig. 3a). These linear scaling laws make intuitive sense for $|Ca_\alpha| \lesssim 1$, as the only forcing in the system is the active stress, which is proportional to $Ca_\alpha$, the fluid dynamics are governed by the linear Stokes equations, and nonlinear effects due to drop deformation are small. Interestingly, the maximum velocity is usually higher when the drop breaks symmetry ($w_i = 1$), except in certain cases at large values of $|Ca_\alpha|$, where the drop undergoes large deformations (e.g., $w_i = 0$ and $Ca_\alpha \gtrsim 5$).

What is the energy expenditure of the active drop? We compute the total entropy production rate as follows:

$$\rho T \dot{s}_{tot} = \rho T \int_A dA \dot{s} = \int_A dA \sigma : \nabla u, \quad (9)$$

where $\rho$ is the fluid density, $T$ is the temperature, and $\dot{s} = \partial_t s + u \cdot \nabla s$. The total entropy production can be decomposed into two

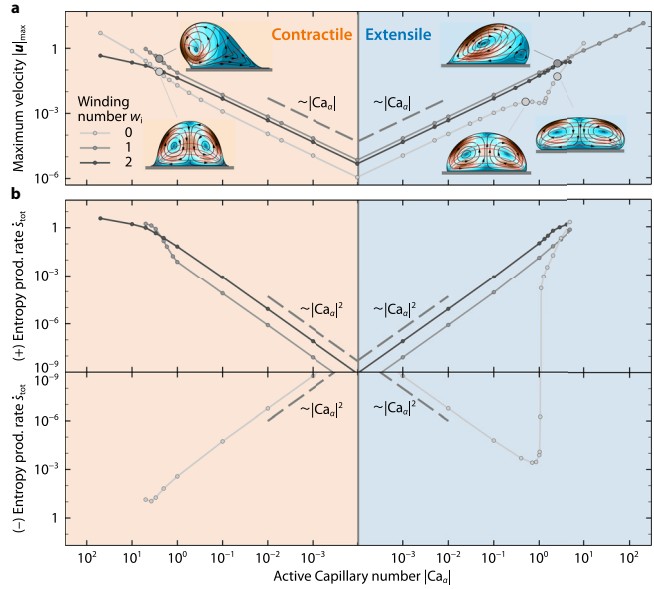

**Fig. 3 | Flow and dissipation inside surface-attached active drops follow simple scaling laws. a** Maximum velocity $|u|_{max}$ inside the active drop as a function of the active Capillary number $Ca_\alpha$ for contractile (left) and extensile (right) active stresses, and different values of the interfacial winding number $w_i$, and $w_s = 0$. **b** Total entropy production rate $\dot{s}_{tot}$ of the active drop as a function of $Ca_\alpha$, under the same conditions as in (**a**).

contributions, i.e., the dissipation associated with viscous stresses, and the work done by the active units:

$$\dot{s}_v = \int_A dA\,\boldsymbol{\sigma}_v : \boldsymbol{\nabla u}, \quad \dot{s}_a = \int_A dA\,\boldsymbol{\sigma}_a : \boldsymbol{\nabla u}, \tag{10}$$

where $\boldsymbol{\sigma}_v = \mu[\boldsymbol{\nabla u} + (\boldsymbol{\nabla u})^T]$ and $\boldsymbol{\sigma}_a = -\alpha \boldsymbol{pp}$ are the viscous and active components of the total stress tensor, respectively. We compute Eqs. (9 and 10) for the case where units are tangentially aligned with the substrate, $w_s = 0$. Viscous dissipation is always irreversible, increasing the system's entropy, $\dot{s}_v > 0$. In our system, we find that the dimensionless viscous dissipation scales as $\dot{s}_v \sim Ca_\alpha^2$ for all values of $w_i$ when $|Ca_\alpha| \lesssim 1$, and is maximum when symmetry is broken, i.e., $w_i = 1$. Conversely, the active contribution scales as $\dot{s}_a \sim -Ca_\alpha^2$ for all values of $w_i$ when $|Ca_\alpha| \lesssim 1$. Interestingly, only when $w_i = 0$ does viscous dissipation remain smaller than the work done by active forces to sustain the self-generated flow and ordering, implying a negative value of $\dot{s}_{tot}$ (Fig. 3b, bottom). This behavior reverses only in the extensile case, $Ca_\alpha > 0$, when the drop undergoes large deformations—and the flow inside the drop consequently changes its handedness—i.e., when $Ca_\alpha \gtrsim 1$, at which point $\dot{s}_{tot}$ becomes positive (Fig. 3b, top right).

### Reversible shape and flow control through time-dependent anchoring of active units

Our results thus far highlight the pivotal importance of interfacial anchoring on drop morphodynamics—suggesting a potential route toward reversibly *controlling* drop shape and flows. To explore this possibility, we vary the anchoring of the active units at the liquid–air interface over time (Fig. 4a, bottom). The upper panel of Fig. 4a shows the initial condition of the drop when the units are tangentially aligned with both the substrate and the liquid–air interface ($w_s = w_i = 0$), and $Ca_\alpha = 2.5$. The anchoring with the substrate remains fixed, while the winding number at the liquid–air interface varies over time, as specified in the bottom panel of Fig. 4b, which also shows the maximum velocity inside the drop, $|\boldsymbol{u}|_{max}$, to determine when the drop reaches steady state. At time $t = 10$, the drop reaches a stable steady-state shape and flow (Fig. 4b). At $t = 11$, we drastically change the interfacial winding number $w_i$ from $w_i = 0$ to $w_i = 1$, forcing the units to orient perpendicular to the liquid–air interface. After a transient period ($11 \lesssim t \lesssim 15$; Fig. 4c), the drop acquires a new steady state (Fig. 4d). To test if this

process is reversible, at $t = 40$, we drastically change the winding number back to $w_i = 0$. Although the transient states corresponding to $w_i = 1 \rightarrow w_i = 0$ (Fig. 1e) are not identical to its opposite, $w_i = 0 \rightarrow w_i = 1$ (Fig. 1c), implying that changing the boundary condition is not equivalent to time reversal, the system relaxes back to the same steady state over time (Fig. 1b). This result shows that this minimal active matter system allows for reversible control of stable steady-state shapes.

## Discussion

Developing principles to control the shape, flow, and interfacial morphodynamics of active matter is key to understanding the functioning of biological systems and designing functional soft active materials. Here, by considering a minimal model of a surface-attached active drop, we have shown that such an active system can exhibit a rich array of steady-state shapes and internal flows. Additionally, we showed that these states can be controlled reversibly, either through surface anchoring of the active components or by varying their activity strength. These features cannot be captured by the thin-film approximation, which is often used to describe such systems.

Despite the rich behaviors found in this active system, we made several simplifications and assumptions in formulating our theoretical description. For example, our model neglects hydrodynamic inertial effects, proliferation, and complex fluid rheology—features that can be present in both synthetic and living systems. Similarly, here we neglected density variations of active components, as well as complex chemo–mechanical interactions with the rigid substrate, which are known to play a crucial role in the wetting of biomolecular condensates[33,58–61]. Moreover, we have only considered planar droplets here—accounting for fully three-dimensional drops may lead to richer drop morphodynamics and hydrodynamic instabilities[28]. Exploring the impact of these additional complexities on the morphodynamics of active systems is a promising avenue for future research.

Altogether, by deepening our understanding of the morphodynamics of active drops, we hope our results will inspire and inform new experiments in active systems. A particularly promising direction is the experimental implementation of time-varying anchoring conditions. One possibility is to use suspensions of microtubules and light-activatable motor proteins, which can organize microtubule structure and flows upon illumination near solid substrates[62–64]. Although this

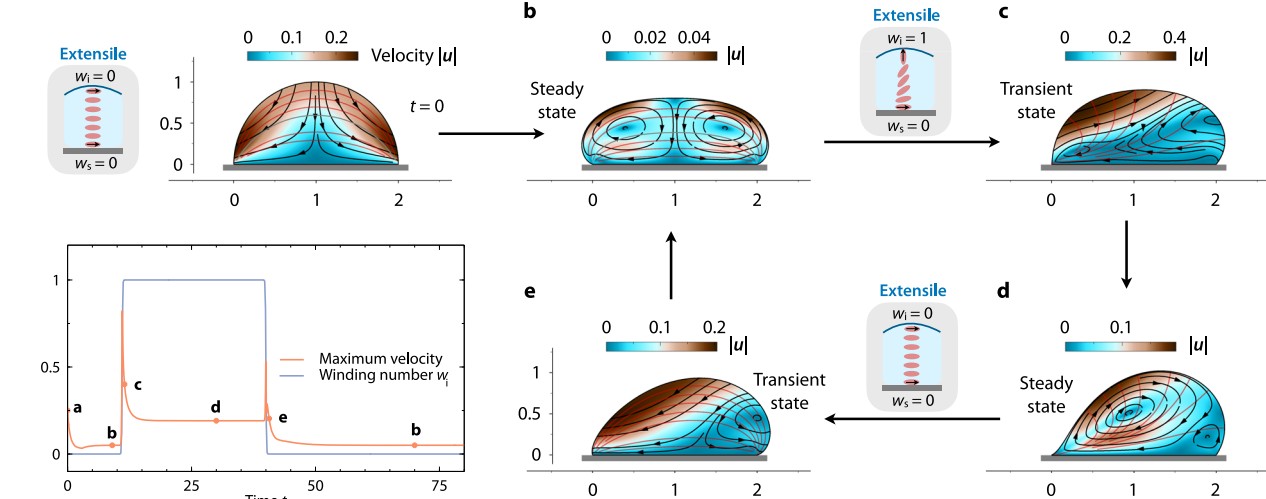

**Fig. 4 | The morphodynamics of surface-attached active drops can be controlled reversibly through the anchoring of active units. a** Initial shape and flow of the drop with $w_s = 0$, $w_i = 0$, and $Ca_\alpha = 2.5$ (top). Interfacial winding number $w_i$ and maximum velocity $|\boldsymbol{u}|_{max}$ as functions of time (bottom). The dots indicate the times shown in (**b**–**e**). **b** Steady state of the drop with $w_s = 0$, $w_i = 0$. **c** Transient state

of the drop immediately after $w_i$ is changed drastically from $w_i = 0$ to $w_i = 1$, at which point the drop begins to break symmetry. **d** After a transient period, the drop reaches steady state with $w_i = 1$. **e** Transient state of the drop after $w_i$ is changed back to $w_i = 0$. After another transient period, the drop achieves the same steady state shown in (**b**).

method may not directly control the anchoring of active units at the solid substrate, it offers a feasible route to modulate microtubule organization and flow at the surface, making it a promising candidate for testing our theoretical predictions in surface-attached droplets. Other avenues include the use of stimuli-responsive surface treatments—such as pH- or photoresponsive polymer brushes and thermoresponsive coatings like PNIPAM—which could enable reversible switching of surface anchoring through environmental cues. Systems with electrically or magnetically responsive active particles, including charged colloids or magnetic bacteria, could also allow for dynamic reorientation near interfaces via external fields. Finally, modulating receptor–ligand interactions through the use of specific surface proteins or local chemical gradients could provide biochemical control over cellular adhesion and orientation at surfaces.

## Data availability
The raw data supporting the findings of this study are available at Zenodo (https://doi.org/10.5281/zenodo.17872650).

## Code availability
The simulation files supporting the findings of this study are available at Zenodo (https://doi.org/10.5281/zenodo.17872650).

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

## Acknowledgements

We thank the members of the Datta Lab for their valuable feedback. A.M.-C. acknowledges support from the Princeton Center for Theoretical Science, the Center for the Physics of Biological Function, and the Human Frontier Science Program through the grant LT000035/2021-C. S.S.D. acknowledges support from NSF Grants CBET-1941716 and DMR-2011750, as well as the Camille Dreyfus Teacher-Scholar and Pew Biomedical Scholars Programs.

## Author contributions

A.M.-C.: conceptualization, methodology, software, validation, formal analysis, investigation, data curation, writing—original draft, writing—review and editing, visualization, project administration, and funding acquisition. S.S.D.: conceptualization, resources, writing—review and editing, supervision, project administration, and funding acquisition.

## Competing interests

The authors declare no competing interests.
