## [Transparent Peer Review file · Nature Communications]

Morphodynamics of surface-attached active drops

Corresponding Author: Dr Alejandro Martinez-Calvo

Version 0:

Reviewer comments:

Reviewer #1

(Remarks to the Author)

This manuscript presents a numerical investigation of the dynamics of surface-attached drops of an active nematic fluid. The model couples the configuration of the nematic constituents (in the strong elastic limit) to the fluid flow they induce, which causes deformations of the drop surface. The focus is on understanding the spontaneous shapes and flows that emerge as a function of the anchoring boundary conditions applied on the nematic on both the flat substrate and drop interface.

Depending on these boundary conditions, a variety of drop shapes are observed, some symmetric and some not. The authors also show how transitions between shapes can be triggered by changing boundary conditions over time.

Overall, the paper is well written and the findings are interesting and a useful addition to the existing literature on such systems. However, I have several comments and questions that the authors should address before I can recommend publication.

Major comments:

1. The system of interest is nematic but relies on a mathematical model that describes the orientation field in terms of a director field $p(x,t)$. It is understood that the two orientations p and $-p$ describe the same nematic field, and indeed the active stress and the relaxation equation (4) are invariant under the transformation $p \rightarrow -p$. However, describing a nematic field using a vector director is not always equivalent to using a nematic Q tensor. In the present work, this is clear from the boundary conditions: for a true nematic, the two winding numbers $w=0$ and 2 describe the same orientation and therefore should not differ. Another place where this is apparent is at the contact line when $w_s=w_i=0$: in that case, a defect is created at the contact line, which is an artefact of using a vector to describe the nematic. One more difference lies in the types of defects that are allowed mathematically: using a vector director does not permit fractional charge defects ($+ or - 1/2$), unlike in a true nematic where these defects are in fact prevalent. This should be at the very least discussed, and the resulting limitations should be acknowledged. In particular, this suggests that some of the findings reported in the paper may not be observed in systems with true nematic symmetry (such as microtubule/kinesin motor suspensions for instance).
2. The authors neglect elastic stresses in the momentum balance, and only account for viscous and active stresses. This is a convenient approximation, especially if the focus is on understanding the role of active stresses on system dynamics. However, this approximation seems inconsistent with the neglect of advective terms in equation (4). The current version of equation (4) assumes strong elasticity (so that advection and rotation by the flow is negligible; however, if elasticity is strong, so are elastic stresses so neglecting them in the momentum balance is questionable. A more rigorous discussion of these approximations is desirable.
3. The two quantities in equation (8) are the rate of viscous dissipation and active power input in the fluid bulk. While it's fine to interpret them as contributions to the rate of change of entropy, another way of looking at them would be to perform a mechanical energy budget starting from the momentum equation (by dotting it with the fluid velocity, integrating over the volume of fluid, and performing integration by parts). This would lead to an energy budget where the sum of viscous dissipation and active power is balanced by surface terms (likely involving surface tractions, that can be related to capillary stresses and to the shape of the drop). This may provide a physical interpretation for the signs reported in figure 3(b) for the net rate of entropy production.
4. On page 5, the authors mention the "disappearance of the point defects at the triple contact point" when extensile stresses become strong. I don't see how that's possible. As mentioned in my point 1 above, the choice of $w_s=w_i=0$ for the surface winding numbers must create a singularity in the director field at the contact point. How can a defect not be present there?

This should be explained. A zoomed-in picture showing the field lines in this region could be helpful to understand what's going on (field lines are difficult to see in the current figures).

Minor comments:

5. The linear and quadratic scalings with capillary number observed in figures 3(a) and (b) are unsurprising, since the only forcing in the system is the active stress and the governing equations for the flow are linear.
6. The idea of controlling drop morphology by varying the anchoring boundary conditions are interesting. Can the authors speculate how this could be achieved in experiments?
7. Can the authors speculate on how their findings might change in three-dimensional drops? What is the contact line was allowed to move?

(Remarks on code availability)

Reviewer #2

(Remarks to the Author)

The present manuscript numerically investigates the steady-states of surface attached active drops. Activity arises from a stress-field (contractile or extensile) related to a nematic order parameter. The authors study the drop shapes and internal velocity profiles depending on a so-called "active capillary number" (an adimensional number which characterizes the strength of the active stress compared to the capillary pressure) and the anchoring conditions of the nematic order parameter at the substrate and the free interface.

The originality of this manuscript lies in the systematic study of two-dimensional droplets of active liquid (aka active liquid ridges) beyond the thin film limit. Here the variation of the director field in the z-direction and the full velocity profiles (in the Stokes limit) are explicitly resolved. The field of the nematic order parameter is treated in the strong elastic limit and is solved adiabatically as a boundary value problem coupled to the drop shape.

Overall the manuscript is clearly laid out and nicely illustrated. My main criticism of this manuscript is the formulation of the boundary conditions for the director field, which are imposing a polarization of the liquid at the boundaries. In particular this choice imposes a (i) symmetry-breaking, where the authors claim a spontaneous symmetry-breaking takes place. (ii) it is not clear to me, how the polarity is treated at the pinned triple contact points of the droplets, which constitute a singular point. The anchoring conditions are crucial to this manuscript. I strongly recommend the authors provide a clearer explanation.

Below is a detailed criticism.

1. I have a major concern with the formulation of the boundary conditions for the order parameter p . While at the substrate ($y=0$) the boundary condition is $p = R_{s,e} \cdot x$, the anchoring condition at the free interface is expressed as $p = R_{i,t}$, with R_i and R_s denoting rotation matrices and t denoting the tangent of the interface. How are the triple contact points treated? Taking as example the initial condition with a semicircular shape, the imposed boundary conditions for p are ill posed and cannot be fulfilled. For example, $\omega = 0$ implies an anchoring at the free surface of $p = e_z$ and $p = -e_z$ at the left and right triple point, respectively (assuming e_z denotes a unit vector orthogonal to the substrate and the tangent vector at the free surface, t , has a direction as indicated in Fig. 1a), while at the same time a substrate anchoring with $\omega = 0$ implies that $p = e_x$ all along the substrate. The authors should state clearly how the triple contact points are treated. On page 5 the authors mention the appearance of point defects in the director field at the triple contact points for the anchoring conditions $\omega_s = \omega_i = 0$. How are these defects consistent with the imposed boundary conditions?

2. Another concern, still related to the anchoring conditions for the order parameter p , is the choice to specify the polar order at the interfaces, instead of only the nematic order. Since the authors refer in their manuscript only to active nematic particles, how can this choice be justified within the same framework? I may be wrong, but the symmetry-breaking observed in Fig. 2a for $\omega_s = 0$ and $\omega_i = 1$ is imposed by the rigid (and to some point pathological) anchoring of p to the two different interfaces and is not a spontaneous process as claimed by the authors on page 5 (left column, first paragraph).

3. As far as I understood the triple contact points are pinned to the substrate. A priori I do not have problem with this limitation. However, the obtained steady-state drop shapes (Fig. 2a) are intriguing. Especially for the depicted combination of $\omega_s = 0$, $\omega_i = 1$ at high contractile stress the droplet contracts strongly close to the left contact point and is only attached via a long flat foot to the right contact point. What does the velocity field and the director field look like in these flat extensions? A supplementary figure which shows the drop profile, velocity and p fields in the vicinity of the triple contact points for the parameter combinations depicted in Fig. 2a is necessary. This could also clarify the impact of the anchoring conditions on the velocity and director field (see concerns raised above).

4. Fig. 2a (bottom row, extensile stress, $Ca_\alpha = 4.5$): The two lobes of the droplet extend into the substrate. Please comment how this is possible? I assume the problem can be resolved by stating that the substrate only extends from $x = 0$ to $x = 2$?

5. In Fig. 3: I am not sure I fully understand the physical relevance of the depicted scaling relations, can the authors make an attempt at explaining the scaling? Also, the curves do not look very smooth, it would be more appropriate to show a symbol at the parameters, where the simulations have been performed. The entropy production rate for extensile stresses and

winding number $\omega_i = 0$ changes abruptly at $Ca_\alpha \approx 1$. At the same moment the linear scaling for u_{\max} breaks down. Please comment. Is this change related to a qualitative change in the drop profile, velocity field, etc? Finally, the steady states shown in Fig. 2a correspond to an active capillary number, where the scaling laws are not valid anymore. Please include a (supplementary) figure with a capillary number where the scaling is valid (e.g. $Ca_\alpha = 0.1$).

6. I find the discussion of previous works (thin film models, e.g. on page 5, right col., first paragraph) a bit hand waving and too simplistic. Also the three-dimensional phase-field model by Tjhung et al. (Nat. Comms. 2015, doi: 10.1038/ncomms6420) should be discussed more thoroughly, since there droplet motion is found for contractile droplets in the absence of self-propulsion. Furthermore, the thin film model by Stegemerten et al. (Soft Matter 2022, doi: 10.1039/d2sm00648k) identifies bistable behavior for droplets with active stresses. I agree that most thin film models (like the one from Stegemerten et al.) do not take into account the z-dependence of the director field. However, there, symmetry is broken spontaneously and not by the imposed surface anchoring.

Minor remarks/typos:

- The use of the term "active suspension" in the abstract (line 4) is misleading since it implies the presence of concentration fluctuations of the suspended particles, which is not considered in the manuscript. "Active liquid" is maybe more appropriate?
- Fig. 2 caption: "... and the grey curves depict the orientation of the active units." Some curves appear grey, but most are red.
- Fig. 3: The color code is misleading. The font colors for "Extensile" and "Contractile" correspond to the colors used for curves with winding number 1 and 2.

(Remarks on code availability)

Version 1:

Reviewer comments:

Reviewer #2

(Remarks to the Author)

The authors have undertaken extensive numerical calculations to investigate i.e. the effect of the (formulation of the) boundary conditions, elastic stresses and advection of the nematic field on the steady state profiles (Figs.~S1 - S3). I feel that all points from my previous report, except for point 2, have been sufficiently addressed in the revised manuscript.

Concerning point 2 from my previous report, I agree with the authors that the bulk equations respect nematic symmetry. However, the boundary conditions do not, since they are explicitly expressed by Dirichlet conditions on the polarity field \mathbf{p} . Furthermore, since the droplet is not self-polarizing, polarization in the bulk only arises from the boundary conditions (free boundary and contact with the substrate).

From the mathematical formulation (Eq.~5b) and line 163 it is clear that the boundary conditions act directly on the polarization and do not respect nematic symmetry. However, this important detail is not sufficiently highlighted in the accompanying text nor is it clearly stated in the new paragraph (lines 338--378, Describing the nematic field using a nematic tensor). The double arrows for \mathbf{p} used in Fig.~1 are misleading. I might be wrong, but my impression is that the loss of symmetry-breaking for the case ($\omega_s=0$, $\omega_i=1$, Fig. S1) can be attributed to the formulation of the boundary conditions respecting nematic symmetry and not to the introduction of the nematic tensor \mathbf{Q} for the bulk equations.

The authors should discuss more openly the origin of symmetry-breaking for the case ($\omega_s=0$, $\omega_i=1$) in the paragraph dedicated to Fig. 2a (line 282 onwards). Furthermore, Fig.~1 should be modified to reflect the true nature of the boundary conditions.

Furthermore I have a technical question concerning Fig. S1a (case $\omega_i=1$). The red lines marking the orientation of the active units indicate the presence of a defect on the free boundary in the center of the droplet at position $x=1$. Is that impression correct? I would have expected that a formulation of the boundary conditions in terms of a Dirichlet condition on \mathbf{Q} with $\tilde{S}=1$ (lines 730--745) does not lead to a defect on the boundary. Please comment or rectify.

Minor points:

Please make sure that all supplementary figures are referenced from the main text, e.g. it seems that Fig. S4 is not referenced from the main text.

Please indicate the winding number at the substrate ω_s in the caption of Fig.~S4.

(Remarks on code availability)

Reviewer #4

(Remarks to the Author)
see attached pdf for the report

(Remarks on code availability)

Version 2:

Reviewer comments:

Reviewer #2

(Remarks to the Author)
The authors have satisfactorily addressed all the points raised in my previous review. The manuscript is now suitable for publication in its present form.

(Remarks on code availability)

Reviewer #4

(Remarks to the Author)
In the revised manuscript, the authors have sufficiently addressed my comments and also the original concerns raised by reviewer #1

(Remarks on code availability)

Reviewer 1 comments:

This manuscript presents a numerical investigation of the dynamics of surface-attached drops of an active nematic fluid. The model couples the configuration of the nematic constituents (in the strong elastic limit) to the fluid flow they induce, which causes deformations of the drop surface. The focus is on understanding the spontaneous shapes and flows that emerge as a function of the anchoring boundary conditions applied on the nematic on both the flat substrate and drop interface. Depending on these boundary conditions, a variety of drop shapes are observed, some symmetric and some not. The authors also show how transitions between shapes can be triggered by changing boundary conditions over time.

Overall, the paper is well written and the findings are interesting and a useful addition to the existing literature on such systems. However, I have several comments and questions that the authors should address before I can recommend publication.

Response: We thank the Reviewer for the time spent reading our paper, and for providing such thoughtful and constructive feedback. It is gratifying that they found our paper to be well written and the findings are interesting and a useful addition to the existing literature on active matter. We also appreciate the Reviewer's suggestion for more rigorous consideration of the problem formulation and boundary conditions, which has helped us better understand this active system and significantly improve the manuscript. We have followed all of their helpful suggestions in revising the manuscript, as detailed in the responses below. In summary, we have:

- Performed new simulations for a true nematic, in which the nematic field is described by a nematic tensor \mathbf{Q} rather than a director vector \mathbf{p} . These simulations show that, although no spontaneous symmetry breaking emerges, the main results obtained using \mathbf{p} remain conceptually equivalent. We have added a new subsection discussing these new results, a Supplementary Information section describing this framework in detail, and a new Supplementary Information figure.
- Performed new simulations showing that nematic advection effects have a minimal impact on the steady-state drop shapes and internal flows, supporting the assumption that considering the strong elastic limit while neglecting elastic stresses is reasonable. We have updated both the Main Text and Supplementary Information to highlight these new findings and have added two new supplementary figures where we consider the advection of the nematic director field \mathbf{p} and the nematic tensor \mathbf{Q} .
- Characterized the flow and nematic orientation near the triple contact points where the drop is pinned, as well as in thin-film regions for certain parameter values, showing that both are accurately resolved. We have added a new supplementary figure showing zoomed-in views of these regions.
- Conducted new simulations in which the drop is not pinned to the solid substrate, by introducing a finite slip length between the active liquid and the substrate to model friction. We show that, under conditions where the active drop breaks symmetry, it can migrate in a treadmill fashion. We have included a new supplementary figure showing this new result.

We believe these new results have improved our work significantly, and we are grateful to the Reviewer for their guidance in obtaining these new results.

Major comments:

1. The system of interest is nematic but relies on a mathematical model that describes the orientation field in terms of a director field $\mathbf{p}(\mathbf{x}, t)$. It is understood that the two orientations \mathbf{p} and $-\mathbf{p}$ describe the same nematic field, and indeed the active stress and the relaxation equation (4) are invariant under the transformation $\mathbf{p} \rightarrow -\mathbf{p}$. However, describing a nematic field using a vector director is not always equivalent to using a nematic \mathbf{Q} tensor. In the present work, this is clear from the boundary conditions: for a true nematic, the two winding number $w = 0$ and 2 describe the same orientation and therefore should not differ. Another place where this is apparent is at the contact line when $w_s = w_i = 0$: in that case, a defect is created at the contact line, which is an artifact of using a vector to describe the nematic. One more difference lies in the types of defects that are allowed mathematically: using a vector director does not permit fractional charge defects ($+ or -1/2$), unlike in a true nematic where these defects are in fact prevalent. This should be at the very least discussed, and the resulting limitations should be acknowledged. In particular, this suggests that some of the findings reported in the paper may not be observed in systems with true nematic symmetry (such as microtubule/kinesin motor suspensions for instance).

Response: We are grateful to the Reviewer for raising this important issue. As the Reviewer points out, describing a nematic through \mathbf{p} is not always equivalent to describing it through a nematic tensor \mathbf{Q} . It is indeed the case of our configuration. To address this issue, we have conducted new numerical simulations in which the nematic is described by \mathbf{Q} . We consider the

following equation for \mathbf{Q} , under the same assumption as in Eq. (4) of the manuscript—nematic order is established much more quickly than the flows generated within the drop:

$$\mathbf{0} = [a - b \text{tr}(\mathbf{Q}^2)] \mathbf{Q} + K_Q \nabla^2 \mathbf{Q}, \quad (1)$$

where $\mathbf{Q} = S (\mathbf{p}\mathbf{p} - \frac{1}{2}\mathbf{I})$, with S a scalar representing the strength of the order parameter, K_Q the Frank elastic constant, and a and b parameters from a Landau expansion in the order parameter that determine the isotropic–nematic transition ($a > 0$ in the nematic phase and $b > 0$ for stability). These parameters define the maximum value the order parameter can take, $S_{\max} = \sqrt{2a/b}$. Using \mathbf{Q} , the active stress tensor becomes $\boldsymbol{\sigma}_a = -\alpha \mathbf{Q}$. To make Eq. (1) dimensionless, we choose $Q_c = \sqrt{a/b}$ as characteristic quantity for the nematic tensor, which yields:

$$\mathbf{0} = [1 - \text{tr}(\tilde{\mathbf{Q}}^2)] \tilde{\mathbf{Q}} + \tilde{K} \nabla^2 \tilde{\mathbf{Q}}, \quad (2)$$

where $\tilde{\mathbf{Q}} = \tilde{S} (\mathbf{p}\mathbf{p} - \frac{1}{2}\mathbf{I})$ is the dimensionless nematic tensor, $\tilde{K} = (K_Q/a)/R^2$ is a dimensionless parameter that compares the elastic coherence length $\ell_Q = \sqrt{K/a}$ and the drop radius R , and $\tilde{S} = S/\sqrt{a/b}$ is the dimensionless strength of the nematic order parameter. Equation (3) of the manuscript describing the fluid stress tensor reads in dimensionless form:

$$\tilde{\boldsymbol{\sigma}} = -\tilde{\Pi} \mathbf{I} + \nabla \tilde{\mathbf{u}} + (\nabla \tilde{\mathbf{u}})^T - \text{Ca}_\alpha \tilde{\mathbf{Q}}, \quad (3)$$

where, in this case, the active Capillary number reads: $\text{Ca}_\alpha \equiv \alpha \sqrt{a/b}/(\gamma/R)$.

To compare with the results of the Main Text solving for \mathbf{p} , we consider both planar and orthogonal anchoring of the nematic with the solid substrate, i.e., $\tilde{\mathbf{Q}} = \tilde{S} (\hat{e}_x \hat{e}_x - \frac{1}{2}\mathbf{I})$ and $\tilde{\mathbf{Q}} = \tilde{S} (\hat{e}_y \hat{e}_y - \frac{1}{2}\mathbf{I})$, respectively. At the liquid–air interface, we impose either tangential anchoring, $\tilde{\mathbf{Q}} = \tilde{S} (\hat{t}\hat{t} - \frac{1}{2}\mathbf{I})$, or orthogonal anchoring, $\tilde{\mathbf{Q}} = \tilde{S} (\hat{n}\hat{n} - \frac{1}{2}\mathbf{I})$, where \hat{t} and \hat{n} are the unit tangential and normal vectors to the liquid–air interface, respectively. For simplicity, in all simulations we set $\tilde{S} = 1$ at the boundaries, and $\tilde{K} = 1$. The figure attached below depicts the results of numerical simulations. Panel a is equivalent to Fig. 2a of the manuscript, corresponding to planar anchoring with the substrate ($w_s = 0$) and both tangential ($w_i = 0$) and orthogonal ($w_i = 1$) anchoring with the liquid–air interface. Panel b is equivalent to Fig. 2b of the manuscript, where we show that the steady states of a surface-attached active drop for planar ($w_s = 0$) and orthogonal ($w_s = 1$) unit anchoring with the substrate are equivalent under the transformations $w \rightarrow w + \pi/2$ and $\text{Ca}_\alpha \rightarrow -\text{Ca}_\alpha$. Here, $|\text{Ca}_\alpha| = 2.5$.

As the Reviewer anticipated, within this framework there are only two anchoring conditions for fixed substrate anchoring ($w_s = 0$ in this case), i.e., $w_i = 0$ and $w_i = 1$. Thus, symmetry breaking is also a consequence of describing the nematic via \mathbf{p} and not considering a true nematic. However, the fact that planar ($w_s = 0$) and orthogonal ($w_s = 1$) substrate anchoring are equivalent under the transformations $w \rightarrow w + \pi/2$ and $\text{Ca}_\alpha \rightarrow -\text{Ca}_\alpha$ is also true when the nematic is described by \mathbf{Q} . We have revised the Main Text of the manuscript to include a careful discussion of both frameworks and the corresponding results (see new subsection in the Main Text, lines 338–370). Additionally, we have added a new section to the Supplementary Information describing the derivation and non-dimensionalization of the equations above. The figure above has also been included in the Supplementary Information.

2. The authors neglect elastic stresses in the momentum balance, and only account for viscous and active stresses. This a convenient approximation, especially if the focus is on understanding the role of active stresses on system dynamics. However, this approximation seems inconsistent with the neglect of advective terms in equation (4). The current version of equation (4) assumes strong elasticity (so that advection and rotation by the flow is negligible; however, if elasticity is strong, so are elastic stresses so neglecting them in the momentum balance is questionable. A more rigorous discussion of these approximations is desirable.

Response: The Reviewer raises a very important issue. In the original manuscript, we considered the strong elastic limit for the director field \mathbf{p} but neglected nematic elastic stresses in the momentum conservation equation. To address the role of nematic hydrodynamic advection and elastic relaxation, we conducted simulations in which the strong elastic limit is relaxed—that is, we included nematic advective terms, and the only stresses acting on the fluid are active, with no elastic contributions. If we describe the nematic through the tensor $\tilde{\mathbf{Q}}$, Eq. (2) now reads:

$$\tilde{\Gamma}_Q \left[\partial_t \tilde{\mathbf{Q}} + \tilde{\mathbf{u}} \cdot \nabla \tilde{\mathbf{Q}} - (\tilde{\mathbf{Q}} \cdot \tilde{\boldsymbol{\Omega}} - \tilde{\boldsymbol{\Omega}} \cdot \tilde{\mathbf{Q}}) \right] = \left[1 - \text{tr}(\tilde{\mathbf{Q}}^2) \right] \tilde{\mathbf{Q}} + \tilde{K}_Q \nabla^2 \tilde{\mathbf{Q}}, \quad (4)$$

where $2\tilde{\boldsymbol{\Omega}} = \nabla \tilde{\mathbf{u}} - (\nabla \tilde{\mathbf{u}})^T$ is the dimensionless vorticity tensor, and $\tilde{\Gamma}_Q \equiv (\Gamma_Q/\mu)\gamma/(aR)$ is a dimensionless parameter that compares the nematic rotational viscosity Γ_Q with the liquid viscosity μ . We conducted new numerical simulations solving Eq. (4), considering $\tilde{\Gamma}_Q = 1$. While the transient dynamics with nematic advection differ from those in the strong-elastic limit, the final steady-state shapes and flows that the drop adopts are nearly identical in both cases, as shown by the figure attached below.

We conduct the same numerical calculations but now describing the nematic through the director field \mathbf{p} , as in the original manuscript, where now Eq. (4) of the manuscript reads:

$$\tilde{\Gamma} \left(\partial_t \mathbf{p} + \tilde{\mathbf{u}} \cdot \nabla \mathbf{p} + \tilde{\boldsymbol{\Omega}} \cdot \mathbf{p} \right) = \nabla^2 \mathbf{p}, \quad (5)$$

where $\tilde{\Gamma} \equiv (\Gamma R^2/K)/t_c = (\Gamma R^2/K)/(\mu R/\gamma)$, is a dimensionless parameter that compares the characteristic elastic relaxation time scale of the nematic with the viscous characteristic time scale of the liquid $t_c = \mu R/\gamma$. Similarly to the case of Eq. (4) of this rebuttal, here the transient dynamics considering \mathbf{p} advection are also different compared to the strong-elastic limit, but the final steady-state shapes are identical in both cases, as shown in the figure attached below.

We have included the two figures above as supplementary figures, added supplementary sections describing these results in depth, and included a new sentence summarizing the main findings in the revised Main Text (lines 370-378).

3. The two quantities in equation (8) are the rate of viscous dissipation and active power input in the fluid bulk. While it's fine to interpret them as contributions to the rate of change of entropy, another way of looking at them would be to perform a mechanical energy budget starting from the momentum equation (by dotting it with the fluid velocity, integrating over the volume of fluid, and performing integration by parts). This would lead to an energy budget where the sum of viscous dissipation and active power is balanced by surface terms (likely involving surface tractions, that can be related to capillary stresses and to the shape of the drop). This may provide a physical interpretation for the signs reported in figure 3b for the net rate of entropy production.

Response: We thank the Reviewer for this thoughtful suggestion. We indeed keep track of the rate of change of energy associated with surface tension as a function of time, which reads:

$$\partial_t E_\gamma = - \int_A dA \gamma \mathbf{u} \cdot \mathbf{n} \nabla \cdot \mathbf{n}. \quad (6)$$

However, Fig. 3 of the manuscript only reports contributions to the energy budget at steady state, where the velocity at the interface of the drop is zero, implying $\partial_t E_\gamma = 0$, and therefore only viscous dissipation rate and active power contribute to the rate of change of the energy budget. Nonetheless, to better understand some of the changes in entropy production rate and $|\mathbf{u}|_{\max}$ beyond the linear regime, we have included new steady-state snapshots in Fig. 3. For example, the abrupt change at $Ca_a \approx 1$ for $w_i = 0$ is a consequence of the significant change in droplet shape as capillary pressure becomes comparable to active stresses. In this specific case, the handedness of the two vortices reverses due to the shape change, which causes the entropy production rate to change sign (see new sentence in lines 417-418).

4. On page 5, the authors mention the “disappearance of the point defects at the triple contact point” when extensile stresses become strong. I don't see how that's possible. As mentioned in my point 1 above, the choice of $w_s = w_i = 0$ for the surface winding numbers must create a singularity in the director field at the contact point. How can a defect not be present there? This should be explained. A zoomed-in picture showing the field lines in this region could be helpful to understand what's going on

(field lines are difficult to see in the current figures).

Response: We thank the Reviewer for identifying this issue and apologize for the oversight. The Reviewer is correct—the defects at the triple contact point persist throughout the simulation, as shown in the zoomed-in view below. We have revised that sentence in the Main Text of the manuscript (lines 276-277) and included the figure below as a supplementary figure, which shows zoomed-in views of the triple contact points and the thin-film regions for the cases $(w_i, Ca_\alpha) = (0, -4.5), (0, 4.5),$ and $(1, -4.5)$.

Minor comments:

5. The linear and quadratic scalings with capillary number observed in figures 3a and b are unsurprising, since the only forcing in the system is the active stress and the governing equations for the flow are linear.

Response: We agree with the Reviewer. To emphasize it, we have included the following sentence in corresponding section of the revised manuscript (lines 387-391): *These linear scaling laws make intuitive sense for $|Ca_\alpha| \lesssim 1$, as the only forcing in the system is the active stress, which is proportional to Ca_α , the fluid dynamics are governed by the linear Stokes equations, and nonlinear effects due to drop deformation are small.*

6. The idea of controlling drop morphology by varying the anchoring boundary conditions are interesting. Can the authors speculate how this could be achieved in experiments?

Response: We thank the Reviewer for raising this important point. Potential ways to control the anchoring conditions over time in experiments include:

- Using suspensions of microtubules and light-activatable motor proteins. These motor proteins can crosslink and organize the microtubules into different structures upon illumination, thereby enabling temporal control (see Refs. [1–3]). While this approach may not directly control anchoring, it can modulate the microtubule structure and flow at the surface, making it a promising candidate for testing our ideas in surface-attached droplets.
- Using stimuli-responsive surface treatments (e.g., pH-responsive polymer brushes or photoresponsive molecules grafted to substrates) which could allow for reversible switching between different anchoring conditions via chemical or light stimuli. Temperature-responsive polymers like PNIPAM could also provide thermal control over surface interactions.
- For systems with electrically/magnetically responsive active particles (e.g., charged colloids, magnetic bacteria), external fields could dynamically reorient particles near interfaces.
- Receptor-ligand interactions to control cells e.g., using different surface proteins or local chemical gradients to modulate bacterial adhesion and orientation at surfaces.

We have added a few sentences to the Discussion section (lines 478-498) highlighting how some of these approaches could be used to implement the proposed time-varying anchoring conditions.

7. Can the authors speculate on how their findings might change in three-dimensional drops? What if the contact line was allowed to move?

Response: We thank the Reviewer for raising these interesting ideas. Considering three-dimensional drops introduces additional possibilities for the orientation of active units along the now two-dimensional substrate and liquid–air interface, as well as the inclusion of new terms in the free energy functional [4], both of which are likely to result in richer drop morphodynamics. One intriguing possibility is the emergence of non-axisymmetric shapes in 3D, or, for instance, Rayleigh-Plateau instabilities when the droplet is extruded from of the substrate, as in cases where $(w_s, w_i) = (0, 1)$ and $\text{Ca}_\alpha = -4.5$ in both \mathbf{Q} and \mathbf{p} descriptions (Figs. 2 and S1). We have added a sentence in the Discussion section to highlight that this is an interesting direction for future research (lines 472-475).

To examine how our results change when the contact line is allowed to move, we explore this possibility by considering the following Navier slip boundary condition at the solid substrate:

$$\mathbf{u} = \ell_s \hat{\mathbf{e}}_y \cdot [\nabla \mathbf{u} + (\nabla \mathbf{u})^T] \cdot (\mathbf{I} - \hat{\mathbf{e}}_u \hat{\mathbf{e}}_y) = \ell_s \partial_y u \hat{\mathbf{e}}_x, \quad (7)$$

where ℓ_s denotes the local slip length. We have performed new numerical simulations incorporating this boundary condition for one representative case in which the nematic is described by the director field \mathbf{p} . This condition is $(w_s, w_i) = (0, 1)$ and $\text{Ca}_\alpha = 4.5$. As expected, the drop can migrate only when symmetry is broken, and it does so in a treadmilling fashion. We have included the figure attached below as a supplementary figure in the revised manuscript, and added a new section in the Supplementary Information describing how we model the substrate slip condition.

- [1] T.D. Ross, H.J. Lee, Z. Qu, R.A. Banks, R. Phillips, and M. Thomson, “Controlling organization and forces in active matter through optically defined boundaries,” *Nature* **572**, 224–229 (2019).
- [2] R. Zhang, S.A. Redford, P.V. Ruijgrok, N. Kumar, A. Mozaffari, S. Zemsky, A.R. Dinner, V. Vitelli, Z. Bryant, M.L. Gardel, *et al.*, “Spatiotemporal control of liquid crystal structure and dynamics through activity patterning,” *Nat. Mater.* **20**, 875–882 (2021).
- [3] F. Yang, S. Liu, H.J. Lee, R. Phillips, and M. Thomson, “Dynamic flow control through active matter programming language,” *Nat. Mater.*, 1–11 (2025).
- [4] M. C. Marchetti, J.-F. Joanny, S. Ramaswamy, T. B. Liverpool, J. Prost, M. Rao, and R. A. Simha, “Hydrodynamics of soft active matter,” *Rev. Mod. Phys.* **85**, 1143 (2013).
- [5] F. Stegemerten, K. John, and U. Thiele, “Symmetry-breaking, motion and bistability of active drops through polarization-surface coupling,” *Soft Matter* **18**, 5823–5832 (2022).
- [6] M. Ben Amar and L. J. Cummings, “Fingering instabilities in driven thin nematic films,” *Phys. Fluids* **13**, 1160–1166 (2001).
- [7] J. F. Joanny and S. Ramaswamy, “A drop of active matter,” *J. Fluid Mech.* **705**, 46–57 (2012).
- [8] A. Loisy, J. Eggers, and T. B. Liverpool, “Tractionless self-propulsion of active drops,” *Phys. Rev. Lett.* **123**, 248006 (2019).
- [9] S. Trinschek, F. Stegemerten, K. John, and U. Thiele, “Thin-film modeling of resting and moving active droplets,” *Phys. Rev. E* **101**, 062802 (2020).

- [10] A. Loisy, J. Eggers, and T. B. Liverpool, “How many ways a cell can move: the modes of self-propulsion of an active drop,” *Soft Matter* **16**, 3106–3124 (2020).
- [11] S. Shankar, V. Raju, and L. Mahadevan, “Optimal transport and control of active drops,” *Proc. Nat. Acad. Sci. U.S.A.* **119**, e2121985119 (2022).
- [12] A. Ioritim-Uba, A. Loisy, S. Henkes, and T. B. Liverpool, “The nonlinear motion of cells subject to external forces,” *Soft Matter* **18**, 9008–9016 (2022).
- [13] H.Z. Ford, G.L. Celora, E.R. Westbrook, M.P. Dalwadi, B.J. Walker, H. Baumann, C.J. Weijer, P. Pearce, and J.R. Chubb, “Pattern formation along signaling gradients driven by active droplet behaviour of cell groups,” *bioRxiv*, 2024–04 (2024).

Reviewer 2 comments:

The present manuscript numerically investigates the steady-states of surface attached active drops. Activity arises from a stress-field (contractile or extensile) related to a nematic order parameter. The authors study the drop shapes and internal velocity profiles depending on a so-called “active capillary number” (an adimensional number which characterizes the strength of the active stress compared to the capillary pressure) and the anchoring conditions of the nematic order parameter at the substrate and the free interface. The originality of this manuscript lies in the systematic study of two-dimensional droplets of active liquid (aka active liquid ridges) beyond the thin film limit. Here the variation of the director field in the z -direction and the full velocity profiles (in the Stokes limit) are explicitly resolved. The field of the nematic order parameter is treated in the strong elastic limit and is solved adiabatically as a boundary value problem coupled to the drop shape.

Overall the manuscript is clearly laid out and nicely illustrated. My main criticism of this manuscript is the formulation of the boundary conditions for the director field, which are imposing a polarization of the liquid at the boundaries. In particular this choice imposes a (i) symmetry-breaking, where the authors claim a spontaneous symmetry-breaking takes place. (ii) it is not clear to me, how the polarity is treated at the pinned triple contact points of the droplets, which constitute a singular point. The anchoring conditions are crucial to this manuscript. I strongly recommend the authors provide a clearer explanation.

Response: We thank the Reviewer for the time spent reading our paper, and for providing such thoughtful and constructive feedback. It is gratifying that they found our paper to be clearly laid out and nicely illustrated. We also appreciate the Reviewer’s suggestion for more rigorous consideration of the problem boundary conditions, which has helped us better understand this active system and significantly improve the manuscript. We have followed all of their helpful suggestions in revising the manuscript, as detailed in the responses below. In summary, we have:

- Performed new simulations for a true nematic, in which the nematic field is described by a nematic tensor \mathbf{Q} rather than a director vector \mathbf{p} . These simulations show that, although no spontaneous symmetry breaking emerges, the main results obtained using \mathbf{p} remain conceptually equivalent. We have added a new subsection discussing these new results, a Supplementary Information section describing this framework in detail, and a new Supplementary Information figure.
- Performed new simulations showing that nematic advection effects have a minimal impact on the steady-state drop shapes and internal flows, supporting the assumption that considering the strong elastic limit while neglecting elastic stresses is reasonable. We have updated both the Main Text and Supplementary Information to highlight these new findings and have added two new supplementary figures where we consider the advection of the nematic director field \mathbf{p} and the nematic tensor \mathbf{Q} .
- Characterized the flow and nematic orientation near the triple contact points where the drop is pinned, as well as in thin-film regions for certain parameter values, showing that both are accurately resolved. We have added a new supplementary figure showing zoomed-in views of these regions.
- Conducted new simulations in which the drop is not pinned to the solid substrate, by introducing a finite slip length between the active liquid and the substrate to model friction. We show that, under conditions where the active drop breaks symmetry, it can migrate in a treadmilling fashion. We have included a new supplementary figure showing this new result.

We believe these new results have improved our work significantly, and we are grateful to the Reviewer for their guidance in obtaining these new results.

1. I have a major concern with the formulation of the boundary conditions for the order parameter \mathbf{p} . While at the substrate ($y = 0$) the boundary condition is $\mathbf{p} = \mathbf{R}_s \cdot \hat{\mathbf{e}}_x$, the anchoring condition at the free interface is expressed as $\mathbf{p} = \mathbf{R}_i \cdot \mathbf{t}$, with \mathbf{R}_i and \mathbf{R}_s denoting rotation matrices and \mathbf{t} denoting the tangent of the interface. How are the triple contact points treated? Taking as example the initial condition with a semicircular shape, the imposed boundary conditions for \mathbf{p} are ill posed and cannot be fulfilled. For example, $\omega_i = 0$ implies an anchoring at the free surface of $\mathbf{p} = \hat{\mathbf{e}}_z$ and $\mathbf{p} = -\hat{\mathbf{e}}_z$ at the left and right triple point, respectively (assuming $\hat{\mathbf{e}}_z$ denotes a unit vector orthogonal to the substrate and the tangent vector at the free surface, \mathbf{t} , has a direction as indicated in Fig. 1a), while at the same time a substrate anchoring with $\omega_s = 0$ implies that $\mathbf{p} = \hat{\mathbf{e}}_x$ all along the substrate. The authors should state clearly how the triple contact points are treated. On page 5 the authors mention the appearance of point defects in the director field at the triple contact points for the anchoring conditions $\omega_s = \omega_i = 0$. How are these defects consistent with the imposed boundary conditions?

2. Another concern, still related to the anchoring conditions for the order parameter \mathbf{p} , is the choice to specify the polar order at the interfaces, instead of only the nematic order. Since the authors refer in their manuscript only to active nematic particles,

how can this choice be justified within the same framework? I may be wrong, but the symmetry-breaking observed in Fig. 2a for $\omega_s = 0$ and $\omega_i = 1$ is imposed by the rigid (and to some point pathological) anchoring of \mathbf{p} to the two different interfaces and is not a spontaneous process as claimed by the authors on page 5 (left column, first paragraph).

Response: We thank the Reviewer for raising these important points, which we appreciate having the opportunity to clarify. We fully agree with the Reviewer that at the triple contact points, the substrate and interface boundary conditions cannot be simultaneously satisfied in a continuous manner. Our model handles this incompatibility by allowing topological defects to naturally emerge at these locations. Indeed, the resolution of this apparent inconsistency is that when $w_s = w_i = 0$, the boundary conditions for \mathbf{p} to accommodate such surface orientations introduce a defect at the contact line at all times—i.e., this combination of winding numbers results in a singularity in the nematic director field \mathbf{p} where the director field becomes undefined. But we emphasize that this defect at the pinned contact line is perfectly consistent with the imposed boundary conditions for \mathbf{p} . These boundary conditions are imposed to the equations for both components of the director field, $\nabla^2 p_x = 0$, and $\nabla^2 p_y = 0$, where $\mathbf{p} = (p_x, p_y)$.

An important aspect to emphasize here is that \mathbf{p} represents a nematic director field, that is, we describe the active nematic using a vector director. However, the system is not polar: the two orientations \mathbf{p} and $-\mathbf{p}$ represent the same physical state, as both the active stress and the relaxation equation [Eqs. (2) and (4) in the manuscript] are invariant under the transformation $\mathbf{p} \rightarrow -\mathbf{p}$. That said, describing a nematic field using the vector \mathbf{p} is not always equivalent to using the nematic tensor \mathbf{Q} .

To better clarify the differences between both frameworks, we have conducted new numerical simulations in which the nematic is described by \mathbf{Q} . We consider the following equation for \mathbf{Q} , under the same assumption as in Eq. (4) of the manuscript—nematic order is established much more quickly than the flows generated within the drop:

$$0 = [a - b \text{tr}(\mathbf{Q}^2)] \mathbf{Q} + K_Q \nabla^2 \mathbf{Q}, \quad (8)$$

where $\mathbf{Q} = S(\mathbf{p}\mathbf{p} - \frac{1}{2}\mathbf{I})$, with S a scalar representing the strength of the order parameter, K_Q the Frank elastic constant, and a and b parameters from a Landau expansion in the order parameter that determine the isotropic–nematic transition ($a > 0$ in the nematic phase and $b > 0$ for stability). These parameters define the maximum value the order parameter can take, $S_{\max} = \sqrt{2a/b}$. Using \mathbf{Q} , the active stress tensor becomes $\sigma_a = -\alpha\mathbf{Q}$. To make Eq. (8) dimensionless, we choose $Q_c = \sqrt{a/b}$ as characteristic quantity for the nematic tensor, which yields:

$$0 = [1 - \text{tr}(\tilde{\mathbf{Q}}^2)] \tilde{\mathbf{Q}} + \tilde{K} \nabla^2 \tilde{\mathbf{Q}}, \quad (9)$$

where $\tilde{\mathbf{Q}} = \tilde{S}(\mathbf{p}\mathbf{p} - \frac{1}{2}\mathbf{I})$ is the dimensionless nematic tensor, $\tilde{K} = (K_Q/a)/R^2$ is a dimensionless parameter that compares the elastic coherence length $\ell_Q = \sqrt{K_Q/a}$ and the drop radius R , and $\tilde{S} = S/\sqrt{a/b}$ is the dimensionless strength of the nematic order parameter. Equation (3) of the manuscript describing the fluid stress tensor reads in dimensionless form:

$$\tilde{\sigma} = -\tilde{\Pi}\mathbf{I} + \nabla\tilde{\mathbf{u}} + (\nabla\tilde{\mathbf{u}})^T - \text{Ca}_\alpha\tilde{\mathbf{Q}}, \quad (10)$$

where, in this case, the active Capillary number reads: $\text{Ca}_\alpha \equiv \alpha\sqrt{a/b}/(\gamma/R)$.

To compare with the results of the Main Text solving for \mathbf{p} , we consider both planar and orthogonal anchoring of the nematic with the solid substrate, i.e., $\tilde{\mathbf{Q}} = \tilde{S}(\hat{e}_x\hat{e}_x - \frac{1}{2}\mathbf{I})$ and $\tilde{\mathbf{Q}} = \tilde{S}(\hat{e}_y\hat{e}_y - \frac{1}{2}\mathbf{I})$, respectively. At the liquid–air interface, we impose either tangential anchoring, $\tilde{\mathbf{Q}} = \tilde{S}(\hat{t}\hat{t} - \frac{1}{2}\mathbf{I})$, or orthogonal anchoring, $\tilde{\mathbf{Q}} = \tilde{S}(\hat{n}\hat{n} - \frac{1}{2}\mathbf{I})$, where \hat{t} and \hat{n} are the unit tangential and normal vectors to the liquid–air interface, respectively. For simplicity, in all simulations we set $\tilde{S} = 1$ at the boundaries, and $\tilde{K} = 1$. The figure attached below depicts the results of numerical simulations. Panel a is equivalent to Fig. 2a of the manuscript, corresponding to planar anchoring with the substrate ($w_s = 0$) and both tangential ($w_i = 0$) and orthogonal ($w_i = 1$) anchoring with the liquid–air interface. Panel b is equivalent to Fig. 2b of the manuscript, where we show that the steady states of a surface-attached active drop for planar ($w_s = 0$) and orthogonal ($w_s = 1$) unit anchoring with the substrate are equivalent under the transformations $w \rightarrow w + \pi/2$ and $\text{Ca}_\alpha \rightarrow -\text{Ca}_\alpha$. Here, $|\text{Ca}_\alpha| = 2.5$.

The figure above shows that, within this framework, there are only two anchoring conditions for a fixed substrate anchoring ($w_s = 0$ in this case), i.e., $w_i = 0$ and $w_i = 1$. Hence, while symmetry breaking is spontaneous and not the result of pathological boundary conditions for \mathbf{p} , it arises as a consequence of describing the nematic using a director vector field \mathbf{p} . However, the fact that planar ($w_s = 0$) and orthogonal ($w_s = 1$) substrate anchoring are equivalent under the transformations $w \rightarrow w + \pi/2$ and $Ca_\alpha \rightarrow -Ca_\alpha$ is also true when the nematic is described by \mathbf{Q} . Moreover, the figure above shows that for $w_s = 0$ and $w_i = 0$, the same defect in the nematic orientation appears at the triple contact point as when the orientation is described by \mathbf{p} , due to the imposed orientations at the substrate and liquid–air interface. We have revised the Main Text of the manuscript to include a careful discussion of both frameworks and the corresponding results (see new subsection in the Main Text, lines 338–378). Additionally, we have added a new section to the Supplementary Information describing the derivation and non-dimensionalization of the equations above. The figure above has also been included in the Supplementary Information. We appreciate the Reviewer’s thoughtful questions, and have now explicitly clarified all these points in the revised manuscript, guided by their feedback.

3. As far as I understood the triple contacts points are pinned to the substrate. A priori I do not have problem with this limitation. However, the obtained steady-state drop shapes (Fig. 2a) are intriguing. Especially for the depicted combination of $w_s = 0$, $w_i = 1$ at high contractile stress the droplet contracts strongly close to the left contact point and is only attached via a long flat foot to the right contact point. What does the velocity field and the director field look like in these flat extensions? A supplementary figure which shows the drop profile, velocity and \mathbf{p} fields in the vicinity of the triple contact points for the parameter combinations depicted in Fig. 2a is necessary. This could also clarify the impact of the anchoring conditions on the velocity and director field (see concerns raised above).

Response: We thank the Reviewer and apologize for not having carefully characterized and visualized the velocity and director fields near the contact point. The reviewer is also correct regarding the boundary conditions—the contact line is pinned to the substrate, as specified by the boundary condition described on line 160 of the manuscript: $\mathbf{u} = \mathbf{0}$ at $y = 0$. Following their advice, we have included the figure below as supplementary figure, which shows zoomed-in views of the triple contact points and the thin-film regions for the cases $(w_i, Ca_\alpha) = (0, -4.5)$, $(0, 4.5)$, and $(1, -4.5)$. Moreover, we have explicitly mentioned that the drop is *pinned* to the substrate in line 101 of the revised manuscript.

4. Fig. 2a (bottom row, extensile stress, $Ca_\alpha = 4.5$): The two lobes of the droplet extend into the substrate. Please comment how this is possible? I assume the problem can be resolved by stating that the substrate only extends from $x = 0$ to $x = 2$?

Response: We thank the Reviewer for bringing up this important point and apologize for not being clear in the original manuscript regarding the extension of the substrate. The Reviewer is correct, the substrate only extends from $x = 0$ to $x = 2$, and therefore the liquid–air interface does not interact with the solid substrate at any point during the simulation. To clarify this issue, we have explicitly mentioned that the substrate only extends from $0 \leq x \leq 2R$ in lines 101-102 of the revised manuscript.

5. In Fig. 3: I am not sure I fully understand the physical relevance of the depicted scaling relations, can the authors make an attempt at explaining the scaling? Also, the curves do not look very smooth, it would be more appropriate to show a symbol at the parameters, where the simulations have been performed. The entropy production rate for extensile stresses and winding number $\omega_i = 0$ changes abruptly at $Ca_\alpha \approx 1$. At the same moment the linear scaling for $|u|_{\max}$ breaks down. Please comment. Is this change related to a qualitative change in the drop profile, velocity field, etc? Finally, the steady states shown in Fig. 2a correspond to an active capillary number, where the scaling laws are not valid anymore. Please include a (supplementary) figure with a capillary number where the scaling is valid (e.g., $Ca_\alpha = 0.1$).

Response: We thank the Reviewer for bringing up these points. First, the linear scaling of the velocity arises from the fact that the only forcing in the system is the active stress, and the governing Stokes equations for the flow are linear. This scaling holds only when the drop does not deform significantly—that is, at sufficiently small values of the active Capillary number. At higher values of Ca_α and thus larger deformations, nonlinearities introduced by a non-semispherical shape cause the breakdown of the linear scaling. We have revised the main text of the manuscript to clarify these points (see lines 387-391).

Regarding the abrupt change in entropy production rate and $|u|_{\max}$ at $Ca_\alpha \approx 1$ for $\omega_i = 0$ —this is a consequence of the significant change in droplet shape as capillary pressure becomes comparable to active stresses. In this specific case, the handedness of the two vortices reverses due to the shape change, which causes the entropy production rate to change sign. To clarify this point, we have included a new steady-state snapshot in Fig. 3 for $Ca_\alpha = 0.5$ and added a corresponding sentence in the revised manuscript (lines 417-418).

We have also followed the Reviewer’s advice and added symbols to the curves to indicate the parameter values at which the simulations were conducted. Additionally, we have conducted more simulations to make curves smoother.

6. I find the discussion of previous works (thin film models, e.g., on page 5, right col., first paragraph) a bit hand waving and too simplistic. Also the three-dimensional phase-field model by Tjhung *et al.* (Nat. Comms. 2015, doi: 10.1038/ncomms6420) should be discussed more thoroughly, since droplet motion is found for contractile droplets in the absence of self-propulsion. Furthermore, the thin film model by Stegemerten *et al.* (Soft Matter 2022, doi: 10.1039/d2sm00648k) identifies bistable behavior for droplets with active stresses. I agree that most thin film models (like the one from Stegemerten *et al.*) do not take into account the z -dependence of the director field. However, there, symmetry is broken spontaneously and not by the imposed surface anchoring.

Response: We thank the Reviewer for raising this important point and apologize for not providing a sufficiently thorough discussion of previous studies, especially the work by Tjhung *et al.* (Nat. Commun. 2015). We have revised the Discussion section to include a more balanced consideration of related models (lines 472–475).

Regarding the work of Stegemerten *et al.* (Stegemerten *et al.* Soft Matter, 2022 [5]) and similar thin-film models of surface-attached active nematic drops (e.g., [5–13]), it is the opposite—the nematic director is allowed to vary only in the vertical y direction and is assumed to remain uniform along the substrate (x direction). This assumption is central to the thin-film (or lubrication) approximation: owing to the slenderness of the drop, gradients of \mathbf{p} and $\mathbf{u} \cdot \hat{\mathbf{e}}_x$ in the vertical direction dominate, and thus only these terms are retained, with errors of order $O[(\partial_x h)^2]$, where h is the height of the drop. Consequently, symmetry breaking also arises due to surface anchoring in the thin-film approximation (see e.g., [8]), which makes sense, since the thin-film equation is derived from the full equations considered in our work. However, the strong assumptions of the thin-film approximation lead to notable differences with respect to the complete equations when $\partial_x h \sim 1$. For example, due to the assumption of \mathbf{p} being x -independent, when the nematic units are tangentially aligned with both the substrate and the free interface ($w_s = w_i = 0$), no flow is generated inside the drop and it remains undeformed, unlike in the full model. Additionally, in the thin-film framework, the droplet shape is identical for both $w_i = 1$ and $w_i = 2$, despite differences in internal flow and nematic organization [8]. This outcome is an artifact of neglecting variations along x and the vertical component of the flow $\mathbf{u} \cdot \hat{\mathbf{e}}_y$, and depth-averaging to obtain an equation for h . We extensively discussed the limitations of the thin-film approximation and compare thin-film predictions (e.g., from [8]) with results from the full equations in several parts of the original manuscript (see e.g., section *Surface-attached active drops reach stable steady states*, *Surface-attached active drops adopt a rich array of equilibrium shapes and flows*, or lines 193–205 of the revised manuscript).

Regarding the work by Tjhung *et al.* (Nat. Comms. 2015), one of the key differences from our study is that they consider an active polar suspension, rather than an active nematic suspension. Although we cited this work in the original manuscript,

we did not discuss it in detail due to this distinction. Moreover, they considered additional complexities that we disregarded, such as effective friction with the substrate introduced via a finite slip length, which leads to drop self-propulsion. In our case, introducing a slip condition at the solid substrate similarly results in drop self-propulsion (as previously found in the thin-film limit, e.g., [8]). We explore this effect in the full equations by considering the following Navier slip boundary condition at the solid substrate:

$$\mathbf{u} = \ell_s \hat{\mathbf{e}}_y \cdot [\nabla \mathbf{u} + (\nabla \mathbf{u})^T] \cdot (\mathbf{I} - \hat{\mathbf{e}}_u \hat{\mathbf{e}}_y) = \ell_s \partial_y u \hat{\mathbf{e}}_x, \quad (11)$$

where ℓ_s denotes the local slip length. We have performed new numerical simulations incorporating this boundary condition for one representative case in which the nematic is described by the director field \mathbf{p} . This condition is $(w_s, w_i) = (0, 1)$ and $\text{Ca}_\alpha = 4.5$. As expected, the drop can migrate only when symmetry is broken, and it does so in a treadmilling fashion, reaching a constant traveling-wave velocity. We have included the figure attached below as a supplementary figure in the revised manuscript, and added a new section in the Supplementary Information describing how we model the substrate slip condition.

In addition, Tjhung *et al.* (Nat. Comms. 2015) also consider elastic stresses, fluid inertia both inside and outside the drop, and advection of the polar field, all of which we disregarded in our work. In the original manuscript, we considered the strong elastic limit for the director field \mathbf{p} but neglected nematic elastic stresses in the momentum conservation equation. To address the role of nematic hydrodynamic advection and elastic relaxation, we conducted simulations in which the strong elastic limit is relaxed—that is, we included nematic advective terms, and the only stresses acting on the fluid are active, with no elastic contributions. If we describe the nematic through the tensor $\tilde{\mathbf{Q}}$, Eq. (9) now reads:

$$\tilde{\Gamma}_Q \left[\partial_t \tilde{\mathbf{Q}} + \tilde{\mathbf{u}} \cdot \nabla \tilde{\mathbf{Q}} - (\tilde{\mathbf{Q}} \cdot \tilde{\mathbf{\Omega}} - \tilde{\mathbf{\Omega}} \cdot \tilde{\mathbf{Q}}) \right] = \left[1 - \text{tr}(\tilde{\mathbf{Q}}^2) \right] \tilde{\mathbf{Q}} + \tilde{K}_Q \nabla^2 \tilde{\mathbf{Q}}, \quad (12)$$

where $2\tilde{\mathbf{\Omega}} = \nabla \tilde{\mathbf{u}} - (\nabla \tilde{\mathbf{u}})^T$ is the dimensionless vorticity tensor, and $\tilde{\Gamma}_Q \equiv (\Gamma_Q/\mu)\gamma/(aR)$ is a dimensionless parameter that compares the nematic rotational viscosity Γ_Q with the liquid viscosity μ . We conducted new numerical simulations solving Eq. (12), considering $\tilde{\Gamma}_Q = 1$. While the transient dynamics with nematic advection differ from those in the strong-elastic limit, the final steady-state shapes and flows that the drop adopts are nearly identical in both cases, as shown by the figure attached below.

We conduct the same calculations but now describing the nematic through the director field \mathbf{p} , as in the original manuscript, where now Eq. (4) of the manuscript reads:

$$\tilde{\Gamma} \left(\partial_t \mathbf{p} + \tilde{\mathbf{u}} \cdot \nabla \mathbf{p} + \tilde{\mathbf{\Omega}} \cdot \mathbf{p} \right) = \nabla^2 \mathbf{p}, \quad (13)$$

where $\tilde{\Gamma} \equiv (\Gamma R^2/K)/t_c = (\Gamma R^2/K)/(\mu R/\gamma)$, is a dimensionless parameter that compares the characteristic elastic relaxational time scale of the nematic with the viscous characteristic time scale of the liquid $t_c = \mu R/\gamma$. Similarly to the case of Eq. (12) of this rebuttal, here the transient dynamics considering \mathbf{p} advection are also different compared to the strong-elastic limit, but the final steady-state shapes are identical in both cases, as shown in the figure attached below.

We have included the two figures above as supplementary figures, added supplementary sections describing these results in depth, and included a new sentence summarizing the main findings in the revised Main Text (lines 370-378).

Minor remarks/typos:

7. The use of the term “active suspension” in the abstract (line 4) is misleading since it implies the presence of concentration fluctuations of the suspended particles, which is not considered in the manuscript. “Active liquid” is maybe more appropriate?

Response: We agree with the Reviewer and have replaced the term “active suspension” with “active fluid” in the abstract.

8. Fig. 2 caption: “... and the grey curves depict the orientation of the active units.” Some curves appear grey, but most are red.

Response: We thank the Reviewer for pointing out the typo, which we have corrected in the revised manuscript.

9. Fig. 3: The color code is misleading. The font colors for “Extensile” and “Contractile” correspond to the colors used for curves with winding number 1 and 2.

Response: We thank the Reviewer for pointing out the confusion caused by the color scheme. We have revised Fig. 3 using a black-and-white color palette for clarity.

-
- [1] T.D. Ross, H.J. Lee, Z. Qu, R.A. Banks, R. Phillips, and M. Thomson, “Controlling organization and forces in active matter through optically defined boundaries,” *Nature* **572**, 224–229 (2019).
 - [2] R. Zhang, S.A. Redford, P.V. Ruijgrok, N. Kumar, A. Mozaffari, S. Zemsky, A.R. Dinner, V. Vitelli, Z. Bryant, M.L. Gardel, *et al.*, “Spatiotemporal control of liquid crystal structure and dynamics through activity patterning,” *Nat. Mater.* **20**, 875–882 (2021).
 - [3] F. Yang, S. Liu, H.J. Lee, R. Phillips, and M. Thomson, “Dynamic flow control through active matter programming language,” *Nat. Mater.* , 1–11 (2025).
 - [4] M. C. Marchetti, J.-F. Joanny, S. Ramaswamy, T. B. Liverpool, J. Prost, M. Rao, and R. A. Simha, “Hydrodynamics of soft active matter,” *Rev. Mod. Phys.* **85**, 1143 (2013).
 - [5] F. Stegemerten, K. John, and U. Thiele, “Symmetry-breaking, motion and bistability of active drops through polarization-surface coupling,” *Soft Matter* **18**, 5823–5832 (2022).
 - [6] M. Ben Amar and L. J. Cummings, “Fingering instabilities in driven thin nematic films,” *Phys. Fluids* **13**, 1160–1166 (2001).
 - [7] J. F. Joanny and S. Ramaswamy, “A drop of active matter,” *J. Fluid Mech.* **705**, 46–57 (2012).
 - [8] A. Loisy, J. Eggers, and T. B. Liverpool, “Tractionless self-propulsion of active drops,” *Phys. Rev. Lett.* **123**, 248006 (2019).
 - [9] S. Trinschek, F. Stegemerten, K. John, and U. Thiele, “Thin-film modeling of resting and moving active droplets,” *Phys. Rev. E* **101**, 062802 (2020).
 - [10] A. Loisy, J. Eggers, and T. B. Liverpool, “How many ways a cell can move: the modes of self-propulsion of an active drop,” *Soft Matter* **16**, 3106–3124 (2020).
 - [11] S. Shankar, V. Raju, and L. Mahadevan, “Optimal transport and control of active drops,” *Proc. Nat. Acad. Sci. U.S.A.* **119**, e2121985119 (2022).
 - [12] A. Ioritim-Uba, A. Loisy, S. Henkes, and T. B. Liverpool, “The nonlinear motion of cells subject to external forces,” *Soft Matter* **18**, 9008–9016 (2022).
 - [13] H.Z. Ford, G.L. Celora, E.R. Westbrook, M.P. Dalwadi, B.J. Walker, H. Baumann, C.J. Weijer, P. Pearce, and J.R. Chubb, “Pattern formation along signaling gradients driven by active droplet behaviour of cell groups,” *bioRxiv* , 2024–04 (2024).

Reviewer 2 comments:

The authors have undertaken extensive numerical calculations to investigate i.e., the effect of the (formulation of the) boundary conditions, elastic stresses and advection of the nematic field on the steady state profiles (Figs. S1 - S3). I feel that all points from my previous report, except for point 2, have been sufficiently addressed in the revised manuscript.

Concerning point 2 from my previous report, I agree with the authors that the bulk equations respect nematic symmetry. However, the boundary conditions do not, since they are explicitly expressed by Dirichlet conditions on the polarity field \mathbf{p} . Furthermore, since the droplet is not self-polarizing, polarization in the bulk only arises from the boundary conditions (free boundary and contact with the substrate).

From the mathematical formulation (Eq. 5b) and line 163 it is clear that the boundary conditions act directly on the polarization and do not respect nematic symmetry. However, this important detail is not sufficiently highlighted in the accompanying text nor is it clearly stated in the new paragraph (lines 338–378, Describing the nematic field using a nematic tensor). The double arrows for \mathbf{p} used in Fig. 1 are misleading. I might be wrong, but my impression is that the loss of symmetry-breaking for the case ($\omega_s = 0$, $\omega_i = 1$, Fig. S1) can be attributed to the formulation of the boundary conditions respecting nematic symmetry and not to the introduction of the nematic tensor \mathbf{Q} for the bulk equations. The authors should discuss more openly the origin of symmetry-breaking for the case ($\omega_s = 0$, $\omega_i = 1$) in the paragraph dedicated to Fig. 2a (line 282 onwards). Furthermore, Fig. 1 should be modified to reflect the true nature of the boundary conditions.

Response: We thank the Reviewer for the time spent reading the new version of our paper, and we are pleased that they found all their points were addressed and that the manuscript improved considerably after implementing their suggestions. We also thank them again for providing new, thoughtful, and constructive feedback that has further improved our work. In particular, we agree with the Reviewer that, in the \mathbf{p} -description of the nematic, the active units become polar at the boundaries by virtue of the imposed boundary conditions. We also agree that this specific point was not sufficiently emphasized in the previous version of the manuscript as a necessary condition for symmetry breaking. To address this issue, we have followed all of their helpful suggestions in revising the manuscript, as detailed in the responses below. In summary, we have:

- Revised both the main text and the Supplementary Information to clarify that the results shown in the main text correspond to the \mathbf{p} -description, in which the active units are polarized at the drop boundaries due to the imposed boundary condition (Eq. 5b). These changes are highlighted in blue in lines 70–73, 107–110, Eq. 5, 158–163, 295, 349–356, 379–381, 729–735, and in the Fig. 1 caption.
- Modified Figs. 1, 2, and 4 to highlight the direction of polarization of the active units at the drop boundaries.

Furthermore I have a technical question concerning Fig. S1a (case $\omega_i = 1$). The red lines marking the orientation of the active units indicate the presence of a defect on the free boundary in the center of the droplet at position $x = 1$. Is that impression correct? I would have expected that a formulation of the boundary conditions in terms of a Dirichlet condition on \mathbf{Q} with $\tilde{S} = 1$ (lines 730–745) does not lead to a defect on the boundary. Please comment or rectify.

Response: We thank the Reviewer for this insightful question. The observation is correct—a defect indeed appears at the center of the liquid–air interface. This arises because the imposed boundary conditions enforce the nematic director to be parallel to the substrate and orthogonal to the liquid–air interface. Satisfying both anchoring conditions simultaneously requires the presence of a defect at the interface center to accommodate the resulting mismatch in orientation.

Minor points:

Please make sure that all supplementary figures are referenced from the main text, e.g., it seems that Fig. S4 is not referenced from the main text.

Response: We thank the Reviewer for pointing out this issue. All Supplementary Figures are now referenced either in the Supplementary Information or in the main text (e.g., see lines 847–849).

Please indicate the winding number at the substrate ω_s in the caption of Fig. S4.

Response: We have indicated the substrate winding number ($\omega_s = 0$) in the updated caption of Fig. S4.

We believe that with the improvements we have made, the manuscript is now suitable for publication in *Nature Communications*. Thank you for your consideration.

Reviewer 4 comments:

Reviewer 1 has raised a couple of valid and important points. I believe the authors have adequately addressed most of these comments by extending their model to incorporate advection terms and a \mathbf{Q} -tensor description, as well as by performing additional simulations and analysis. However, I am still confused with the new results which they presented as part of the response to Comment 1 and I think the authors should provide more clarifications to my concerns below.

Nematic liquid crystals can both be described by a director field: $\mathbf{p}(\mathbf{r}, t)$ (using the same notation as in the manuscript) or by a \mathbf{Q} -tensor: $\mathbf{Q}(\mathbf{r}, t)$. The dynamics for \mathbf{p} can be described by the Leslie-Ericksen equation (line 131 in the manuscript), while the dynamics for \mathbf{Q} can be described by the Beris-Edwards equation (line 343 in the manuscript). Far from any topological defects, simulations using either $\mathbf{p}(\mathbf{r}, t)$ or $\mathbf{Q}(\mathbf{r}, t)$ will give identical results, and one can interpret \mathbf{p} as the largest eigenvector of the \mathbf{Q} -tensor.

In the revised manuscript, the authors performed additional simulations using a \mathbf{Q} -tensor description to compare with the original results (done using director field \mathbf{p}). The results for $w_s = 0$ and $w_l = 1$ are qualitatively different for both \mathbf{Q} -tensor description and \mathbf{p} -field description. In particular the result with \mathbf{p} -field has a symmetry-breaking steady state which can cause the droplet to migrate. On the other hand, the steady state solution for \mathbf{Q} -tensor remains symmetric about the central axis. I don't think the authors have explained sufficiently why they have very significant difference. Is it due to the fact that in \mathbf{p} -description, the \mathbf{p} -field can form +1 defects at the contact lines and these defects become unstable in \mathbf{Q} -tensor description?

Response: We thank the Reviewer for the time and effort spent carefully reading our revised manuscript and for acknowledging our extensions to include advection terms, a \mathbf{Q} -tensor formulation, and additional simulations and analyses. We appreciate their thoughtful comments and the opportunity to clarify the origin of the differences between the \mathbf{p} - and \mathbf{Q} -based descriptions.

In the \mathbf{p} -description, the boundary conditions explicitly impose the orientation of the active units at both the solid and liquid–air interfaces, which leads to the units becoming polar at the drop boundaries. A mismatch between these boundary orientations constitutes a necessary condition for symmetry breaking. In contrast, in the \mathbf{Q} -tensor description, the boundary conditions are formulated in terms of the nematic director field and do not prescribe a net polarity at the interfaces. As a result, the \mathbf{Q} -tensor solution remains symmetric about the central axis. The apparent difference between the two formulations therefore arises from the boundary conditions rather than from the instability or suppression of +1 defects in the \mathbf{Q} -tensor case.

We agree with the Reviewer that this point was not discussed or emphasized in the previous version of our manuscript. To clarify this important aspect, we have revised both the main text and the Supplementary Information as follows:

- Revised both the main text and the Supplementary Information to clarify that the results shown in the main text correspond to the \mathbf{p} -description, in which the active units are polarized at the drop boundaries due to the imposed boundary condition (Eq. 5b). These changes are highlighted in blue in lines 70–73, 107–110, Eq. 5, 158–163, 295, 349–356, 379–381, 729–735, and in the Fig. 1 caption.
- Modified Figs. 1, 2, and 4 to highlight the direction of polarization of the active units at the drop boundaries.

We believe that with the improvements we have made, the manuscript is now suitable for publication in *Nature Communications*. Thank you for your consideration.

Reviewer 1 has raised a couple of valid and important points. I believe the authors have adequately addressed most of these comments by extending their model to incorporate advection terms and a \mathbf{Q} -tensor description, as well as by performing additional simulations and analysis. However, I am still confused with the new results which they presented as part of the response to Comment 1 and I think the authors should provide more clarifications to my concerns below.

Nematic liquid crystals can both be described by a director field: $\mathbf{p}(\mathbf{r}, t)$ (using the same notation as in the manuscript) or by a \mathbf{Q} -tensor: $\mathbf{Q}(\mathbf{r}, t)$. The dynamics for \mathbf{p} can be described by the Leslie-Ericksen equation (line 131 in the manuscript), while the dynamics for \mathbf{Q} can be described by the Beris-Edwards equation (line 343 in the manuscript). Far from any topological defects, simulations using either $\mathbf{p}(\mathbf{r}, t)$ or $\mathbf{Q}(\mathbf{r}, t)$ will give identical results, and one can interpret \mathbf{p} as the largest eigenvector of the \mathbf{Q} -tensor.

In the revised manuscript, the authors performed additional simulations using a \mathbf{Q} -tensor description to compare with the original results (done using director field \mathbf{p}). The results for $w_s = 0$ and $w_i = 1$ are qualitatively different for both \mathbf{Q} -tensor description and \mathbf{p} -field description. In particular the result with \mathbf{p} -field has a symmetry-breaking steady state which can cause the droplet to migrate. On the other hand, the steady state solution for \mathbf{Q} -tensor remains symmetric about the central axis. I don't think the authors have explained sufficiently why they have very significant difference. Is it due to the fact that in \mathbf{p} -description, the \mathbf{p} -field can form +1 defects at the contact lines and these defects become unstable in \mathbf{Q} -tensor description?